# Long term O$_3$-precursor relationships in Hong Kong: Field observation and model simulation

Yu Wang[1], Hao Wang[1], Hai Guo*[,1], Xiaopu Lyu[1], Hairong Cheng**[,2], Zhenhao Ling[3], Peter. K.K. Louie[4], Isobel J. Simpson[5], Simone Meinardi[5], Donald R. Blake[5]

[1] Air Quality Studies, Department of Civil and Environmental Engineering, the Hong Kong Polytechnic University, Hong Kong

[2] Department of Environmental Engineering, Wuhan University, Wuhan, China

[3] School of Atmospheric Sciences, Sun Yat-sen University, China

[4] Air Group, Hong Kong Environmental Protection Department, Hong Kong

[5] Department of Chemistry, University of California, Irvine, CA, USA

*Correspondence to*: ceguohai@polyu.edu.hk; chenghr@whu.edu.cn

**Abstract.** Over the past ten years (2005-2014), ground-level O$_3$ in Hong Kong has consistently increased in all seasons except winter, despite the yearly reduction of its precursors, *i.e.*, nitrogen oxides (NO$_x$=NO+NO$_2$), total volatile organic compounds (TVOCs) and carbon monoxide (CO). To explain the contradictory phenomenon, an observation-based box model (OBM) coupled with CB05 mechanism was applied in order to understand the influence of both locally-produced O$_3$ and regional transport. The simulation of locally-produced O$_3$ showed an increasing trend in spring, a decreasing trend in autumn and no changes in summer and winter. The O$_3$ increase in spring was caused by the net effect of more rapid decrease of NO titration and unchanged TVOC reactivity despite decreased TVOC mixing ratios, while the decreased local O$_3$ formation in autumn was mainly due to the reduction of aromatic VOC mixing ratios and the TVOC reactivity and much slower decrease of NO titration. However, the decreased in-situ O$_3$ formation in autumn was overridden by the regional contribution, resulting in elevated O$_3$ observations. Furthermore, the OBM-derived relative incremental reactivity indicated that the O$_3$ formation was VOC-limited in all seasons, and the long-term O$_3$ formation was more sensitive to VOCs and less to NO$_x$ and CO in the past 10 years. In addition, the OBM results found that the contributions of aromatics to O$_3$ formation decreased in all seasons of these years, particularly in autumn, likely due to effective control of solvent-related sources. In contrast, the contributions of alkenes increased, suggesting a continuing need to reduce traffic emissions. The findings provided updated information on photochemical pollution and its impact in Hong Kong.

**Keywords:** Ozone; VOCs; Long term; Observation-based model

# 1 Introduction

Ozone ($O_3$), one of the most important photochemical products influencing atmospheric oxidative capacity, human and vegetation health, and climate change, is formed through a series of photochemical reactions among volatile organic compounds (VOCs) and nitrogen oxides ($NO_x$) in the atmosphere (Seinfeld and Pandis, 2006). Due to the non-linear relationship between $O_3$ and its precursors, the development of appropriate control measures of $O_3$ is still problematic in mega cities (Sillman, 1999).

Distinguished from short-terms $O_3$ studies, investigation of long-term $O_3$ variations enables us to understand the seasonal and inter-annual characteristics of $O_3$, the influence of meteorological parameters on $O_3$ formation, and the $O_3$-precursor relationships in different years. Subsequently, more effective and sustainable $O_3$ control strategies can be formulated and implemented. Hence, earlier efforts have been made to investigate long-term variations of $O_3$ in different atmospheric conditions. For example, multi-year data analysis showed that the $O_3$ levels started to decrease around 2000 in Europe (*e.g.*, Jungfraujoch, Zugspitze, Mace Head) and North America excluding western US rural sites (*e.g.*, US Pacific, Lassen Volcanic National Park) (Lefohn *et al.*, 2010; Cui *et al.*, 2011; Parrish *et al.*, 2012; Pollack *et al.*, 2013; Lin *et al.*, 2017), due to a decrease in the emissions of $O_3$ precursors since the early 1990s (Cui *et al.*, 2011; Derwent *et al.*, 2013). In contrast, the $O_3$ levels in East Asia increased at a rate of 1.0 ppbv $yr^{-1}$ from 1998 to 2006, based on measurements at Mt. Happo, Japan (Parrish *et al.*, 2012; Tanimoto, 2009). In China, with rapid economic growth and urbanization over the past three decades, increasing $O_3$ levels have been found in many locations. For instance, based on the data collected between 1991 and 2006 at Lin'an, a $NO_x$-limited rural area close to Shanghai, Xu *et al.* (2008) reported that the maximum mixing ratios of $O_3$ increased by 2.0% $yr^{-1}$, 2.7% $yr^{-1}$, 2.4% $yr^{-1}$ and 2.0% $yr^{-1}$ in spring, summer, autumn and winter, respectively, which were likely related to increased mixing ratios of $NO_2$. In the North China Plain (NCP), Ding *et al.* (2008) reported that $O_3$ in the lower troposphere over Beijing had a positive trend of ~2.0% $yr^{-1}$ from 1995 to 2005, while Zhang *et al.* (2014) found that the daytime average $O_3$ in summer in Beijing significantly increased by 2.6 ppbv $yr^{-1}$ from 2005 to 2011, due to decreased NO titration (-1.4 ppbv $yr^{-1}$ of $NO_x$ over the study period) and elevated regional background $O_3$ levels (~0.58 - 1.0 ppbv $yr^{-1}$) in the NCP.

Hong Kong, together with the inland Pearl River Delta (PRD) region of southern China, has suffered from high $O_3$ mixing ratios in recent years (Chan *et al.*, 1998a; Chan *et al.*, 1998b; Wang and Kwok, 2003; Ding *et al.*, 2004; Zhang *et al.*, 2007; Guo *et al.*, 2009 and 2013). In 2014, $O_3$ exceeding the Chinese national air quality standard (80 ppbv, for the daily 8 hour maximum average, DMA8) was > 90 days in some areas of the PRD, with the highest DMA8 value of 165 ppbv (GDEMC and HKEPD, 2015). Based on the observational data at a newly-established regional monitoring network, Li *et al.* (2014) found that $O_3$ mixing ratios in the inland PRD region increased at a rate of 0.86 ppbv $yr^{-1}$ from 2006 to 2011 because of the rapid reduction of NO in this VOC-limited region. Similarly, Wang *et al.* (2009) reported a continuous record of increased surface $O_3$ at Hok Tsui (HT), a regional background site in Hong Kong, with a rate of 0.58 ppbv $yr^{-1}$ based on observations conducted from 1994 to 2007, concluding that the increased $NO_2$ column concentration in upwind eastern China might significantly contribute to the increased $O_3$ in Hong Kong. Even so, knowledge gaps still exist on long-term characteristics of $O_3$, long-term $O_3$-precursor

relationships, and the mechanisms for the varying O₃ trends in the PRD region, because of the lack of long-term observations of VOCs in the region, where photochemical O₃ formation is sensitive to VOCs in urban areas, and where the levels of VOCs and NO$_x$ have varied significantly due to more stringent control measures since 2005 (Zhong *et al*., 2013). It is noteworthy that although Xue *et al*. (2014) reported increasing O₃ trends in 2002-2013 in Hong Kong and investigated the roles of VOCs and NO$_x$ in the long-term O₃ variations, only data in autumn were used, which could not provide a consistently full picture of the long-term variations of O₃, VOCs, NO$_x$ and their relationships.

In this study, field measurements and model simulations were combined to characterize the long-term variations of O₃ and its precursors, the variations of locally-produced O₃, and the impact of regional transport in Hong Kong from 2005 to 2014. In addition, the long-term contribution of different VOC groups to the O₃ formation was explored. The findings aim at providing the most updated information on the characteristics of photochemical pollution and its impact in Hong Kong.

## 2 Methodology

### 2.1 Site description

Field measurements were carried out at the Tung Chung (TC) Air Quality Monitoring Station managed by the Hong Kong Environmental Protection Department (HKEPD). The sampling site (22.29 °N, 113.94 °E) is located at about 24 km southwest of downtown Hong Kong and about 3 km south of the Hong Kong International Airport (Figure 1). The elevation of TC is 37.5 m above sea level. It is surrounded by a newly-developed residential town on the northern Lantau Island, and is downwind of urban Hong Kong and the inland PRD region when easterly and northeasterly winds are prevailing (Ou *et al*., 2015). At TC, the prevailing wind varies by seasons, with east winds for spring and autumn, southwest winds for summer and northeast for winter (see Figure S1). The selection of this site for the trend study was due to its downwind location being a good receptor for urbane plume, suffering high O₃ pollution and having the most comprehensive dataset. More detailed description of the TC site can be found in our previous papers (Jiang *et al*., 2010; Cheng *et al*., 2010; Ling *et al*., 2013; Ou *et al*., 2015).

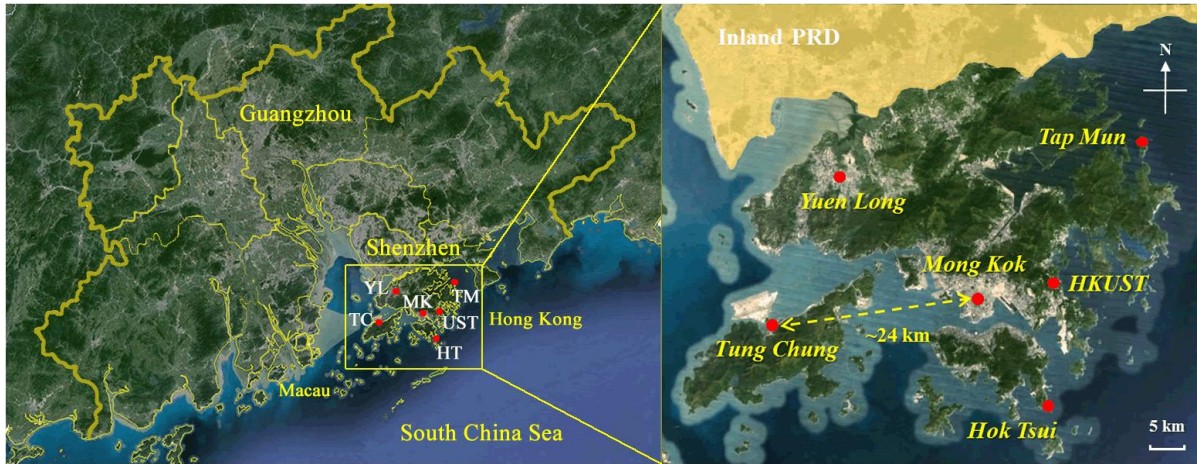

**Figure 1**. Location of the sampling sites and surrounding environments. Guangzhou and Shenzhen are the two biggest cities in the inland PRD region with a population over 10 million for each city. Hok Tsui (HT) and Tap Mun (TM) are regional background sites. The Hong Kong University of Science and Technology (HKUST) is an Air Quality Research Supersite located in suburban area. Yuen Long (YL) is a typical urban site adjacent to main traffic roads and surrounded by residential
and industrial blocks. Mong Kok is a typical roadside site in downtown with high traffic density.

### 2.2 Measurement techniques

Hourly observations of $O_3$, CO, $SO_2$, NO-$NO_2$-$NO_x$ and meteorological parameters at TC from 2005 to 2014 were obtained from the HKEPD (http://epic.epd.gov.hk/ca/uid/airdata). Briefly, $O_3$ was measured using a commercial UV photometric instrument (Advanced Pollution Instrumentation (API), Model 400A) with a detection limit of 0.6 ppbv. CO was measured
with a gas filter correlation CO analyser (Thermo ElectronCorp. (TECO), Model 48C) with a detection limit of 0.04 ppm. $SO_2$ was measured using a pulsed fluorescence analyser (TECO, Model 43A) with a detection limit of 1.0 ppbv. NO-$NO_2$-$NO_x$ were detected using a commercial chemiluminescence with an internal molybdenum converter (API, Model 200A) and a detection limit of 0.4 ppbv. All the time resolutions for these gas analysers are 1 hour. To ensure a high degree of accuracy and precision, the QA/QC procedures for gaseous pollutants were identical to those in the US air quality monitoring program
(http://epic.epd.gov.hk/ca/uid/airdata). The accuracy of the monitoring network is assessed by performance audits, while the precision, a measure of the repeatability, of the measurements is checked in accordance with HKEPD's quality manuals. For the gaseous pollutants, the accuracy and precision within the limits of $\pm 15$ and $\pm 20$ % are adopted, respectively (HKEPD 2015).

Real-time VOC data at TC were also measured by the HKEPD. An online GC-FID analyser (Synspec GC 955, Series 600/800)
was used to collect VOC speciation data continuously with a time resolution of 30 minutes. The VOC analyser consists of two separate systems for detection of $C_2$–$C_5$ and $C_6$–$C_{10}$ hydrocarbons, respectively. Detailed description about the real-time VOC analyser can be found in Lyu *et al*. (2016). There were twenty-eight $C_3$-$C_{10}$ VOC species identified and quantified with this method. In terms of the QA/QC for VOC analysis, built-in computerized programs of quality control systems such as auto-linearization and auto-calibration were used. Weekly calibrations were conducted by using NPL standard gas (National
Physical Laboratory, Teddington, Middlesex, UK). In general, the detection limits of the target VOCs ranged from 2 to 56 pptv. The accuracy of each species measured by online GC-FID was determined by the percentage difference between measured mixing ratio and actual mixing ratio based on weekly span checks and monthly calibrations. The precision was based on the 95% probability limits for the integrated precision check results. The accuracy of the measurements was about 1-7%, depending on the species, and the measurement precision was about 1-10% (Table S1). In addition, the quality of the real-time
data was assured by regular comparison with whole-air canister samples collected and analysed by University of California at Irvine (UCI). More details can be found from previous studies in Hong Kong (Xue *et al*., 2014; Ou *et al*., 2015; Lyu *et al*., 2016).

For data analysis, linear regression and error bars represented as 95% confidence intervals were used. Trends of $O_3$ and its precursors with a *p* value $< 0.05$ were considered significant (Guo *et al*., 2009).

**2.3 Observation-based model**

In this study, an observation-based box model (OBM) coupled with carbon bond mechanism (CB05) was used to simulate
photochemical $O_3$ formation and to evaluate the sensitivity of $O_3$ formation to its precursors. The CB05 mechanism is a condensed mechanism with high computational efficiency and reliable simulation, and has been successfully applied in many emission-based modelling systems such as Weather Research and Forecasting with Chemistry (WRF/Chem) and the Community Multiscale Air Quality (CMAQ) (Yarwood *et al*. 2005; Coates and Butler, 2015). Unlike emission-based models, the OBM in this study is based on the real time observations at the TC site in Hong Kong. The simulation was constrained by
observed hourly data of meteorological parameters (temperature, relative humidity and pressure) and air pollutants (NO, $NO_2$, CO, $SO_2$ and 22 $C_3$-$C_{10}$ VOCs). In the CB05 module, VOCs are grouped according to carbon bond type and the reactions of individual VOCs were condensed using lumped structure technique (conversions from measured VOCs to CB05 grouped species are shown in Table S2). To better describe the photochemical reactions in Hong Kong, the photolysis rates of different species in the OBM model was determined using the output of the Tropospheric Ultraviolet and Visible Radiation model (TUV
v5) (Madronich and Flocke, 1999) based on the actual conditions of Hong Kong, *i.e.*, meteorological parameters, location, and time period of the field campaign. However, it is noteworthy that the atmospheric physical processes (*i.e.*, vertical and horizontal transport), the deposition of species, and the radical loss to aerosol (George *et al.*, 2013; Lakey *et al.*, 2015) were not considered in the OBM model. In addition, a "spin-up" time was not applied in the model to get the radical intermediates well steady which might have caused a slight underestimation on the simulated $O_3$ production (Figure S2) and its sensitivity to
precursors (Figure S3). In this study, we performed day-by-day OBM simulations for 2688 days during 2005−2014, where the missing days were due to lack of real-time VOC data (see Table S3). For each daily simulation, the model was run for a 24-hour period with 00:00 (local time, LT) as the initial time. The model output simulated mixing ratios of $O_3$, radicals (*i.e.*, OH, $HO_2$, RO and $RO_2$) and intermediates. The model performance was evaluated using the index of agreement (IOA) (Huang *et al.*, 2005; Wang *et al.*, 2015; Wang *et al.*, 2013; Lyu *et al.*, 2015).

$$IOA = 1 - \frac{\sum_{i=1}^{n}(O_i - S_i)^2}{\sum_{i=1}^{n}(|O_i - \bar{O}| + |S_i - \bar{O}|)^2} \qquad\qquad \text{(Eq. 1)}$$

where $S_i$ and $O_i$ represent simulated and observed values, respectively, $\bar{O}$ represents the mean of observed values, and *n* is the number of samples. The IOA value lies between 0 and 1. The better agreement between simulated results and observed data, the higher the IOA (Huang *et al.*, 2005).

Apart from the OBM (CB05), which is mainly for condensed VOC groups, a Master Chemical Mechanism (MCM, v3.2) was
applied to inter-compare the modelling performance of OBM (CB05) (shown in section 3.2). Since the MCM utilizes the near-explicit mechanism describing the degradation of 143 primary VOCs and contains around 16,500 reactions involving

5,900 chemical species, it has a better performance in calculating the contribution of individual VOCs to $O_3$ production (Jenkin *et al*., 1997 and 2003; Saunders *et al*., 2003). The hourly input data of meteorological parameters, air pollutants and the photolysis rates in MCM were the same as in CB05. A more detailed description of the MCM can refer to Jenkin *et al*. (1997 and 2003) and Saunders *et al*. (2003). Some developments on localization of the MCM for Hong Kong and addition of

5 chemical reaction pathways of more biogenic VOC species and alkyl nitrates are given in our previous papers (Lam *et al*., 2013; Cheng *et al*., 2013; Ling *et al*., 2014; Lyu et al., 2015).

The measured precursors (*i.e*., VOCs, NO and $NO_2$) at TC are a mixture of regional background values augmented by local source influences, and the two parts are very difficult to be fully separated. It is worth noting that the regional background values are those observed at locations where there is little influence from urban sources of pollution, while the baseline values

mentioned in Section 3.2 are observations made at a site when it is not influenced by recent, locally emitted or produced pollution (TF HTAP, 2010). To minimize the influence of regional transport from the inland PRD region, the real-time regional background values in this study were simply subtracted off from the observations at TC. Previous studies have reported that Tap Mun (TM, 22.47$^o$ N, 114.36$^o$ E) and Hok Tsui (HT, 22.217$^o$ N, 114.25$^o$ E), are two background sites of Hong Kong (Lyu *et al*., 2016; So and Wang, 2003 and 2004; Wang *et al*., 2005; Wang *et al*., 2009). TM is a rural site that is upwind

of Hong Kong in autumn/winter seasons and HT is a background site at southeastern tip of Hong Kong. Good trace gas correlations were found between both sites (Lyu *et al*., 2016). Since not all the data during the entire 10-year period were available at one background site, the hourly measured VOCs at HT and $NO_2$ at TM were treated as background values. The background data were excluded using the equations:

$[VOC]_{local} = [VOC]_{observed} - [VOC]_{background}$ (Eq. 2)

$[NO]_{local} = [NO]_{observed} - [NO]_{background}$ (Eq. 3)

$[NO_2]_{local} = [NO_2]_{observed} - [NO_2]_{background}$ (Eq. 4)

where $[xx]_{local}$, $[xx]_{observed}$, and $[xx]_{background}$ represent the local, observed and background values, respectively. In this study, mixing ratios of 21 anthropogenic VOC species with relatively long lifetimes (5h – 14d) at HT were selected as the background values for deducting from the observed data at TC. The lifetimes of these VOCs were estimated based on the reactions with OH

radicals (Simpson *et al.*, 2010). The rate constants used were from Atkinson and Arey (2003) by assuming a 12-h daytime average OH radical concentration of $2.0 \times 10^6$ molecules cm$^{-3}$. Isoprene was considered as not having a regional impact due to its short lifetime (1-2 h) (Ling *et al*., 2011). Furthermore, the lifetime of $NO_2$ is determined by the main sinks of OH+$NO_2$ reaction and the hydrolysis of $N_2O_5$ at the surface of wet aerosols, which highly depends on meteorological conditions, such as temperature and humidity (Dils *et al*. 2008; Evans and Jacob, 2005). Previous experimental studies showed an exponential

relationship between the $NO_2$ lifetime and temperature (Dils *et al.,* 2008; Merlaud *et al.*, 2011; Rivera *et al.*, 2013), which was used to estimate the lifetime of $NO_2$ in this study. The lifetime of $NO_2$ was calculated to be approximately 3.4$\pm$0.3 h, 2.2$\pm$0.1 h, 2.8$\pm$0.2 h and 5.2$\pm$0.3 h in spring, summer, autumn and winter, respectively, consistent with the lifetimes of $NO_2$ in different seasons in the PRD region (Beirle *et al*., 2011). Considering the shortest distance between the inland PRD and TC (*i.e.,* from the center of Shenzhen to TC site, ~30 km) and the average wind speeds in different seasons (Ou *et al*., 2015), it would take

approximately 3.4±0.3 h, 4.3±0.3 h, 4.0±0.5 h and 3.7±0.4 h in spring, summer, autumn and winter, respectively, for $NO_2$ originating in the inland PRD to arrive at TC. Hence, although $NO_2$ emitted from the inland PRD is slightly more likely to arrive at TC in winter and spring than in summer and fall, the differences in travel time among the seasons are relatively small and it is difficult to be precise with seasonal average estimates of $NO_2$ lifetime and travel time. We have excluded background $NO_2$ values in spring and winter in this study during model simulations, but we recognize the limitations in these calculations.

In addition to the simulation of $O_3$ formation, the precursor sensitivity of $O_3$ formation was assessed by the OBM using the relative incremental reactivity (RIR) (Cardelino and Chameides, 1995; Lu *et al.*, 2010; Cheng *et al.*, 2010; Ling *et al.*, 2011; Xue *et al.*, 2014). A higher positive RIR of a given precursor means a greater probability that reducing emissions of this precursor will more significantly reduce $O_3$ production. The RIR is defined as the percent change in daytime $O_3$ production per percent change in precursors. The RIR for precursor X is given by:

$$\text{RIR}(X) = \frac{[P^S_{O_3-NO}(X) - P^S_{O_3-NO}(X-\Delta X)]/P^S_{O_3-NO}(X)}{\dfrac{\Delta S(X)}{S(X)}} \qquad \text{(Eq. 5)}$$

X represents a specific precursor (*i.e.*, VOCs, $NO_x$, or CO); the superscript "s" is used to denote the specific site where the measurements were made; $S(X)$ is the measured mixing ratio of species X (ppbv); $\Delta S(X)$ is the hypothetical change in the mixing ratio of X; $P^S_{O_3-NO}(X)$ and $P^S_{O_3-NO}(X - \Delta X)$ represent net $O_3$ production in a base run with original mixing ratios, and in a run with a hypothetical change ($\Delta S(X)$; 10% $S(X)$ in this study) in species X. In both runs, $O_3$ production modulated by NO titration is considered during the evaluation period. The $O_3$ production $P^S_{O_3-NO}$ was calculated by the output parameters of the OBM.

$P^S_{O_3-NO}$ is derived from the difference between $O_3$ gross production rate $G^S_{O_3-NO}$ and $O_3$ destruction rate $D^S_{O_3-NO}$ (Eq. 6). $G^S_{O_3-NO}$ is calculated by the oxidation of NO by $HO_2$ and $RO_2$ (Eq. 7), while $D^S_{O_3-NO}$ is calculated by $O_3$ photolysis, reactions of $O_3$ with OH, $HO_2$ and alkenes, and reaction of $NO_2$ with OH (Eq. 8).

$$P^S_{O_3-NO} = G^S_{O_3-NO} - D^S_{O_3-NO} \qquad \text{(Eq. 6)}$$

$$G^S_{O_3-NO} = k_{HO_2+NO}[HO_2][NO] + \sum k_{RO_{2i}+NO}[RO_{2i}][NO] \qquad \text{(Eq. 7)}$$

$$\begin{aligned} D^S_{O_3-NO} = &k_{HO_2+O_3}[HO_2][O_3] + k_{OH+O_3}[OH][O_3] + k_{o(^1D)+H_2O}[O(^1D)][H_2O] + \\ &k_{OH+NO_2}[OH][NO_2] + k_{OLE+O_3}[\text{alkenes}][O_3] \end{aligned} \qquad \text{(Eq. 8)}$$

In Eq.7 and Eq.8, *k* constants are the rate coefficients of their subscript reactions. Values of radicals and intermediates are simulated by the OBM. Details of the calculation can be found in Ling *et al.* (2014) and Xue *et al.* (2014).

Furthermore, the sensitivities in the OBM model to the uncertainties in initial concentrations of ozone precursors have been examined by running the model with varying $NO_2$ or VOCs initial concentrations in the range of ±95% confidence intervals, respectively. The results demonstrate that the modelled $O_3$ production was more sensitive to $NO_2$ than VOCs, with a percentage variation about ±13% and ±3.9%, respectively (see Table S4, Figure S4 & S5). In addition, the uncertainties associated with removing the background concentrations are also evaluated, suggesting a similar trend for simulated locally $O_3$ production for both approaches (see Figure S6 & Tables S5-S7).

## 3 Results and discussion

### 3.1 Long-term trends of $O_3$ and its precursors

Figure 2 shows trends of monthly-averaged mixing ratios of $O_3$ and its precursors, namely $NO_x$, total VOCs (TVOCs) and CO measured at TC in the past 10 years. The TVOCs were defined as the sum of the 22 VOC species listed in Text S1. Note that not all detected VOCs were included in this study because of high rates of missing data. The limited number of VOC precursors would cause missing of reactivity which was estimated < 30% for total hydrocarbons based on our previous study (Guo *et al.*, 2004). The missing reactivity would increase if carbonyls are considered (Cheng *et al.*, 2010). It was found that both monthly-averaged $O_3$ and monthly maximum $O_3$ increased with a rate of 0.56±0.01 ppbv $yr^{-1}$ ($p<0.01$) and 1.92±0.15 ($p<0.05$), respectively. The monthly maximum $O_3$ level, which was defined as the maximum of DMA8 $O_3$ in one month, increased from about 68 ppbv in 2005 to 86 ppbv in 2014, exceeding the ambient air quality standards in Hong Kong (*i.e.*, 80 ppbv). Besides, the number of days per year (d $yr^{-1}$) that DMA8 exceeded 80 ppbv also increased during 2005-2014 (1.16±0.26 d $yr^{-1}$, $p<0.05$, see Figure S7), indicating increasing $O_3$ pollution in Hong Kong. This finding is consistent with other big cities and regions in the world, such as Beijing (Tang *et al.*, 2009), west plains of Taiwan (Chou, *et al.*, 2006), and Osaka (Itano *et al.*, 2007). The annual average $O_3$ concentration in Hong Kong has increased by 0.56 ppbv $yr^{-1}$ in 2004-2015 which is close to that reported for Osaka (0.6 ppbv $yr^{-1}$) in 1985-2002, and in agreement with Lin *et al.* (2017) who found the annual mean $O_3$ over Hong Kong increased by about 0.5 ppbv $yr^{-1}$ over 2000-2014. In contrast, $NO_x$ and CO significantly decreased with an average rate of -0.71±0.01 ppbv $yr^{-1}$ ($p<0.01$) and -29.4±0.05ppbv $yr^{-1}$ ($p<0.01$), respectively, while TVOCs remained unchanged ($p=0.71$) in these years. The decreasing trends of $NO_x$ and CO, also observed in many other high population industrial urban areas (Geddes *et al.*, 2009; Tang *et al.*, 2009), suggest effective reduction of local emissions from transportation, power plants and other industrial activities (HKEPD, 2016). Unlike $O_3$ and $NO_x$, the trend of TVOCs varied across different areas, for example, increasing in Beijing (Tang *et al.*, 2009), decreasing in Toronto (Geddes *et al.*, 2009) and Taiwan (Chou *et al.*, 2006), while almost remained unchanged ($p>0.05$) in Hong Kong (Figure 2). Although the 10-year TVOCs trend did not change, their levels showed clear inter-annual variations in spring and autumn (Figure 3). Moreover, the long-term trends of individual VOCs, except for BVOC, were different from that of TVOCs (see Figure S18) because many control measures were taken in the last decade, which altered the composition of VOCs in the atmosphere, such as the reduction of toluene by solvent usage control

and the increase of alkanes in Liquefied Petroleum Gas (LPG) in 2005-2013 (Ou *et al.*, 2015) and the decrease of LPG-alkanes in 2013-2014 (Lyu *et al.*, 2016).

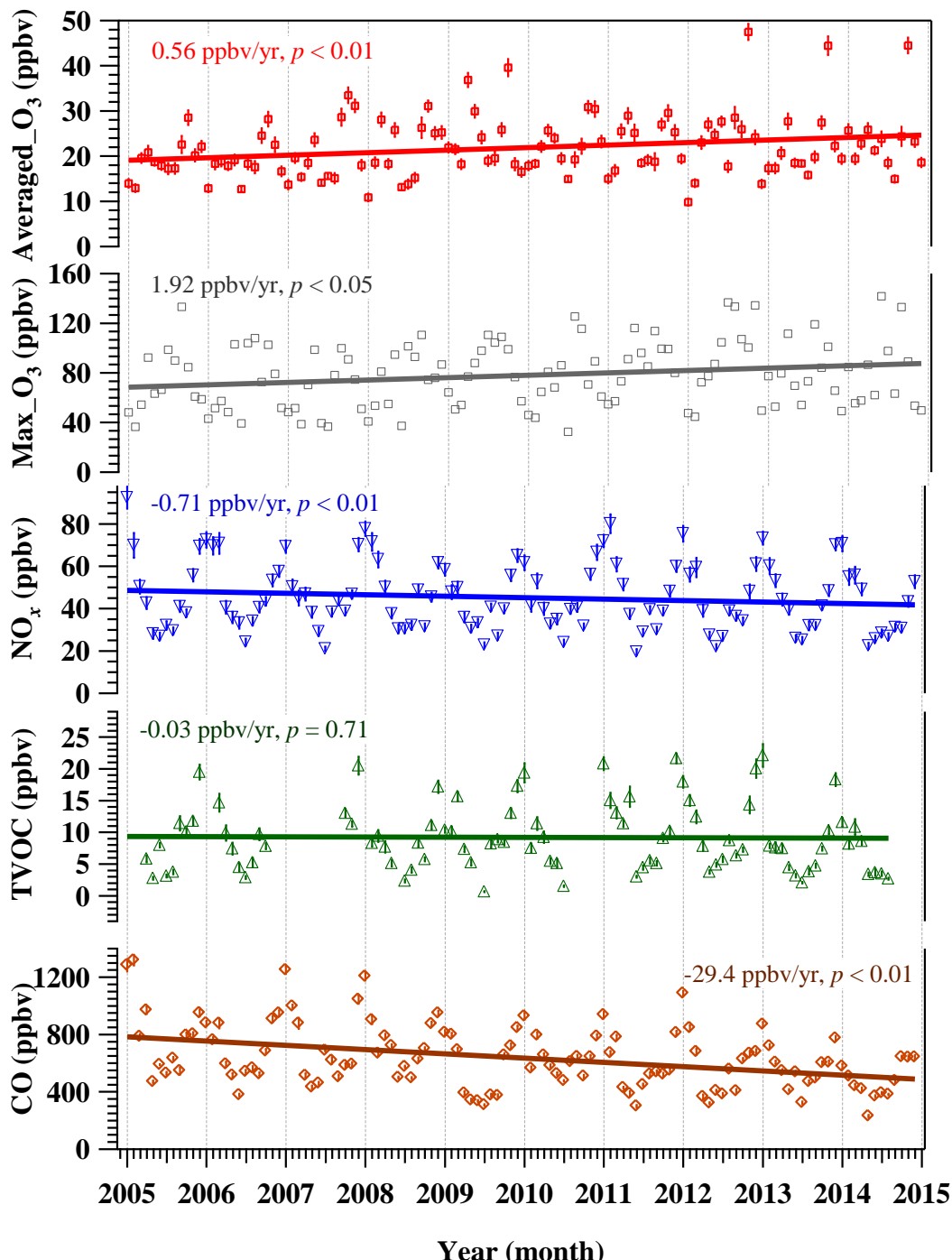

**Figure 2.** Trends of monthly averages of $O_3$ and its precursors, *i.e.*, $NO_x$, TVOCs and CO at TC during 2005–2014. Error bars represent 95% confidence interval of monthly averages.

Figure 3 displays variations of measured $O_3$ and its precursors ($NO_x$, TVOCs and CO) in four seasons in 2005-2014. Here we defined December–February, March–May, June–August and September–November as winter, spring, summer and autumn, respectively. Generally, all precursors showed low values in summer and high levels in winter, mainly due to typical Asian monsoon circulations, which brought in clean marine air in summer and delivered pollutant-laden air from mainland China in winter (Wang *et al*., 2009). Subject to this reason, a similar seasonal variation was observed for the averages of TVOCs at different locations over Hong Kong (see Table S8). With lower (diluted) precursor concentrations, together with high frequency of rainy days, it is not uncommon for Hong Kong to see the lowest $O_3$ values in summertime (see Figure 3).

The long-term trends of CO in all seasons (slopes from spring to winter: $-39.2\pm0.20$, $-16.8\pm0.12$, $-14.4\pm0.16$ and $-50.3\pm0.23$ ppbv $yr^{-1}$, respectively) and $NO_x$ and TVOCs in spring ($NO_x$: $-0.69\pm0.01$ ppbv $yr^{-1}$; TVOCs: $-0.26\pm0.01$ ppbv $yr^{-1}$) and autumn ($NO_x$: $-0.50\pm0.02$ ppbv $yr^{-1}$; TVOCs: $-0.32\pm0.01$ ppbv $yr^{-1}$) showed significant decreases ($p<0.05$), whereas $NO_x$ and TVOCs did not have statistical variations in summer and winter during the 10 years ($p>0.05$). The different inter-annual trends of $NO_x$ and TVOCs in spring/autumn from those in summer/winter were probably because marine air significantly diluted air pollution in summer while continental air masses remarkably burdened air pollution in winter, which concealed the decreased local emissions of $NO_x$ and TVOCs in summer and winter (Wang *et al*., 2009). In contrast, the measured $O_3$ trends significantly increased in spring, summer and autumn, with the rate of $0.51\pm0.05$, $0.50\pm0.04$ and $0.67\pm0.07$ ppbv $yr^{-1}$, respectively ($p<0.05$), while winter $O_3$ levels showed no significant trend ($0.23\pm0.05$ ppbv $yr^{-1}$, $p=0.11$). Apart from the impact of regional transport, the increased spring and autumn $O_3$ in these years was likely due to the reduction of NO titration overrode the $O_3$ decrease owing to the reduction of TVOCs, leading to net $O_3$ increase. Here the NO titration refers to the ''titration reaction'' between NO and $O_3$. Although $NO$–$NO_2$–$O_3$ reaction cycling (including the effects of NO titration, see reactions R1-R3) can be theoretically regarded as a null cycle and provides rapid cycling between NO and $NO_2$, the NO titration effect can retard the accumulation of $O_3$ in an urban environment by means of substantial NO emissions (Chou *et al*., 2006). Indeed, the observed NO at TC site decreased significantly during 2005-2010 (shown in Figure S8), which mitigated the effects of NO titration and led to the increase of $O_3$.

$$NO_2 + hv \rightarrow NO + O \qquad\qquad\qquad\qquad (R1)$$

$$O + O_2 + M \rightarrow O_3 + M \qquad\qquad\qquad\qquad (R2)$$

$$NO + O_3 \rightarrow NO_2 + O_2 \qquad\qquad\qquad\qquad (R3)$$

Interestingly, summer $O_3$ had a significant increase though $NO_x$ and TVOCs showed no differences in these years. Further investigation found that temperature and solar radiation in summer indeed increased ($p<0.05$) in these years (see Figure S9), whereas they had no significant change in other seasons (the reasons remained unclear), consistent with the fact that the increase of temperature and solar radiation would enhance the photochemical reaction rates, partly resulting in $O_3$ increase in

summer (Lee *et al.*, 2014). On the other hand, the unchanged winter $O_3$ trend was in line with the unchanged $NO_x$ and TVOCs values in winter of past 10 years. Again, the impact of regional transport could not be ignored. To better understand the mechanisms of long-term trends of $O_3$ in different seasons in this study, the source origins of $O_3$, *i.e.*, whether it was locally formed or transported from other regions, were explored.

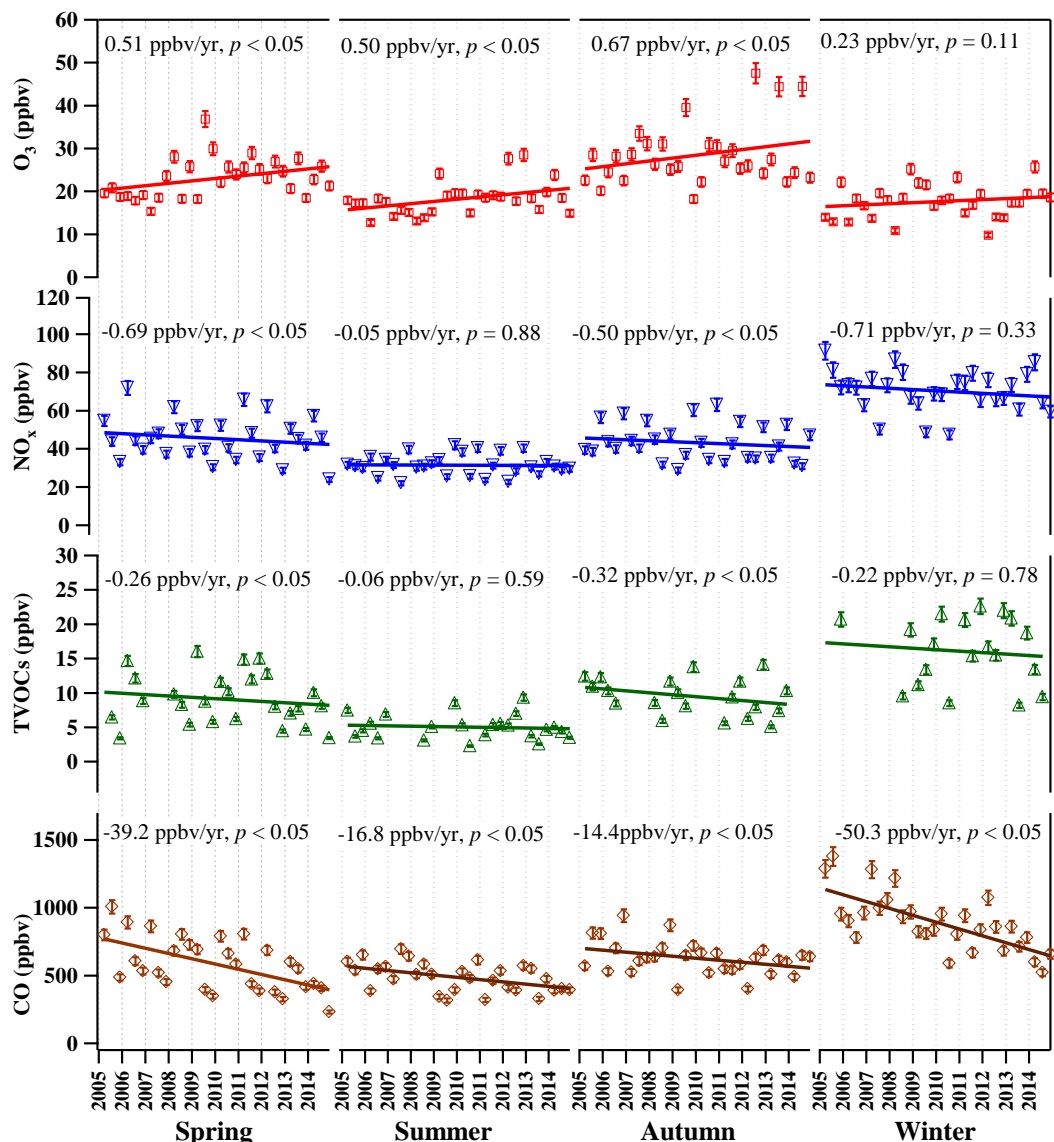

**Figure 3.** Variations of $O_3$ and its precursors in four seasons at TC during 2005–2014. Each data point in the figure is obtained by averaging hourly values into a monthly value. Error bars represent 95% confidence interval of the averages. In the sub-plot for $O_3$ trend in autumn, the three extremely high values are not considered as outliers as each of them represents one month of data with relatively small uncertainty (see Figure S10).

## 3.2 Long-term trends of locally-produced O$_3$ and regional contribution

In this study, the OBM (CB05) model was used to simulate the long-term trends of O$_3$ produced by in-situ photochemical reactions (hereinafter locally-produced O$_3$ (simulated)). The comparisons of simulated and observed O$_3$ at TC during 2005–2014 are shown in Figures S11 (by year) and S12 (by month). Besides, Table S9 lists the IOA values between observed and locally-produced O$_3$ (simulated) at TC site in each year. As shown, the IOA values ranged from 0.71 to 0.89, indicating that the performance of the OBM model on the O$_3$ simulation was acceptable.

It is noteworthy that MCM has better simulation performance than CB05 due to its near-explicit rather than condensed chemical mechanism. Indeed, the overall IOA of MCM modeling (0.89) was higher than that of CB05 (0.81), according to our test on the same sampling days of 2005-2014 (shown in Figure S13, rainy days were excluded). Despite this, the high computational efficiency OBM (CB05) was used for the 10-year day-by-day O$_3$ simulations to investigate the long-term trend of O$_3$ in this study, because the simulated results of both CB05 and MCM models followed similar temporal patterns ($p>0.05$), and the difference of simulated values between the two models was reasonable (IOA value: 0.89 vs. 0.81), revealing that the condensed mechanism of CB05 would not significantly affect the long-term trends of O$_3$ in this study (shown in Figure S14). Previous studies suggest that wind speed of 2 m/s could be used as a threshold to classify regional and local air masses in Hong Kong (Guo *et al*., 2013; Ou *et al*., 2015; Cheung *et al*., 2014). Namely, the O$_3$ values measured with < 2 m/s in this study were considered as locally-produced O$_3$ (hereinafter locally-produced O$_3$ (filtered)). Figure 4 presents the long-term trends of observed daytime O$_3$ (0700-1900LT) and locally-produced O$_3$ (filtered/simulated) in four seasons during 2005-2014. Please note, the trends of observed daytime O$_3$ in the four seasons were consistent with those of 24-hour observed O$_3$ (see Figure S15). It can be seen that the locally-produced O$_3$ (simulated) increased in spring (0.28±0.01 ppbv yr$^{-1}$, $p<0.05$), decreased in autumn (-0.39±0.02 ppbv yr$^{-1}$, $p<0.05$) and showed no change in summer and winter ($p>0.05$). Interestingly, the long-term trend of locally-produced O$_3$ (simulated) in autumn was opposite to that in spring though both NO$_x$ and TVOCs decreased in the two seasons. The reasons were because 1) NO$_x$ decreased faster while TVOCs slower (Figure 3) in spring than in autumn, leading to net increase of O$_3$ formation in spring and decrease in autumn; and 2) TVOC reactivity (described in Text S2 and Table S10) decreased in autumn (-0.03 s$^{-1}$y$^{-1}$, $p<0.05$) but showed insignificant variations in other seasons ($p>0.1$) (Figure S16), resulting in the reduction of O$_3$ production in autumn. The simulated springtime O$_3$ increase and unchanged winter values were consistent with the observed trends, whereas the simulated autumn O$_3$ decrease was opposite to the observed trend for the overall observations. However, locally-produced O$_3$ (filtered) values clearly showed similar trends to locally-produced O$_3$ (simulated) in spring, autumn and winter (see Figure 4), confirming that locally-produced O$_3$ indeed increased in spring and decreased in autumn in these years.

In comparison, a significant difference ($\Delta O_3 = O_3$ overall $_{observed} - O_3$ $_{simulated}$, $p<0.01$) was found between measured and simulated O$_3$ in spring, implying the contribution of regional transport to the measured O$_3$. The 10-year average $\Delta O_3$ was 8.26±1.77 ppbv and the long-term trend of $\Delta O_3$ showed no significant change ($p=0.91$), suggesting that the contribution of regional transport in spring was stable during last decade. The spring pattern of O$_3$ in this study is consistent with the findings

of Li *et al.* (2014) who reported the increasing $O_3$ trend (2.0 ppbv yr$^{-1}$) in spring at urban clusters of PRD from 2006 to 2011. In conclusion, the increasing $O_3$ trend in spring at TC was caused by the increased local $O_3$ production, and the contribution of regional transport was steady in 2005-2014.

Unlike in spring, though the observed and locally-produced $O_3$ (filtered) displayed increasing trends in summer (0.67±0.34 ppbv yr$^{-1}$ and 0.61±0.41 ppbv yr$^{-1}$, respectively; $p<0.05$), locally-produced $O_3$ (simulated) showed no significant change ($p=0.18$), consistent with the unchanged trends of precursors ($NO_x$ and TVOCs) in summer (Figure 3). Note that the influence of annual variation in solar radiation over the 10 years was not considered while the TUV model was used to calculate the photolysis rates, which could mask the actual trends of $O_3$ mixing ratios. Indeed, the total solar radiation (0.24±0.16 MJ m$^{-2}$ yr$^{-1}$, $p<0.01$) and temperature (0.095±0.034 K yr$^{-1}$, $p<0.05$) in summer significantly increased during the past 10 years (see Figure S9), subsequently resulting in the enhanced in-situ photochemical reactivity of VOCs, although their quantitative contributions remain unknown and require further investigation. The increase of solar radiation might be due to the decreasing haze as the air quality has been getting better in Hong Kong and the PRD (Louie *et al.*, 2013). Moreover, the summertime wind speeds significantly decreased at a rate of -0.062±0.041 m·s$^{-1}$yr$^{-1}$ ($p<0.05$), which might accumulate the locally-produced $O_3$. Lastly, the locally-produced $O_3$ (filtered) trend was comparable to observed $O_3$ ($p=0.12$) and locally-produced $O_3$ (simulated) ($p=0.32$), respectively, indicating a negligible impact of regional transport on summer $O_3$ trend in 2005-2014 (thereby $\Delta O_3$ in summer not shown in Figure 4). As such, the increasing trend of summer $O_3$ was partly attributed to the increase of solar radiation and temperature from 2005 to 2014.

Consistent with the decreasing trends of $NO_x$ and TVOCs in autumn, both locally-produced $O_3$ (simulated) ($p<0.05$) and locally-produced $O_3$ (filtered) ($p<0.1$) remarkably decreased, suggesting the dominant impact of VOC reduction over the reduction of NO titration. The decreased locally-produced $O_3$ in autumn was consistent with the results of Xue *et al.* (2014), who found local $O_3$ production decreased in autumn from 2002 to 2013. Furthermore, the 10-year average $\Delta O_3$ was 7.35±3.16 ppbv and the long-term trend of $\Delta O_3$ increased with a rate of 1.09±0.21 ppbv yr$^{-1}$ ($p<0.05$), suggesting an increased contribution of regional transport in autumn during the last decade, in line with the fact that autumn $O_3$ level in inland PRD was higher and increased more rapidly than in Hong Kong (Figure 5), and high $O_3$ mixing ratios were frequently observed in this season due to stronger solar radiation, lower wind speeds, and less vertical dilution of air pollution than in other seasons in this region. In summary, locally-produced $O_3$ in autumn decreased due to the reduction of dominant VOC precursors, while an increased contribution of regional transport negated the local reduction, leading to an elevated $O_3$ observation.

In winter, locally-produced $O_3$ (filtered and simulated) had similar trends ($p=0.93$) and the trends showed no significant changes ($p>0.05$), confirming similar locally-produced $O_3$ in these years, due to insignificant variations of $NO_x$ and TVOCs levels (Figure 3). Besides, locally-produced $O_3$ (both filtered and simulated) presented significant difference from the observed $O_3$ ($p<0.01$), implying the regional contribution in winter. The 10-year average $\Delta O_3$ was 4.56±0.78 ppbv and the long-term trend of $\Delta O_3$ was not significant ($p=0.98$).

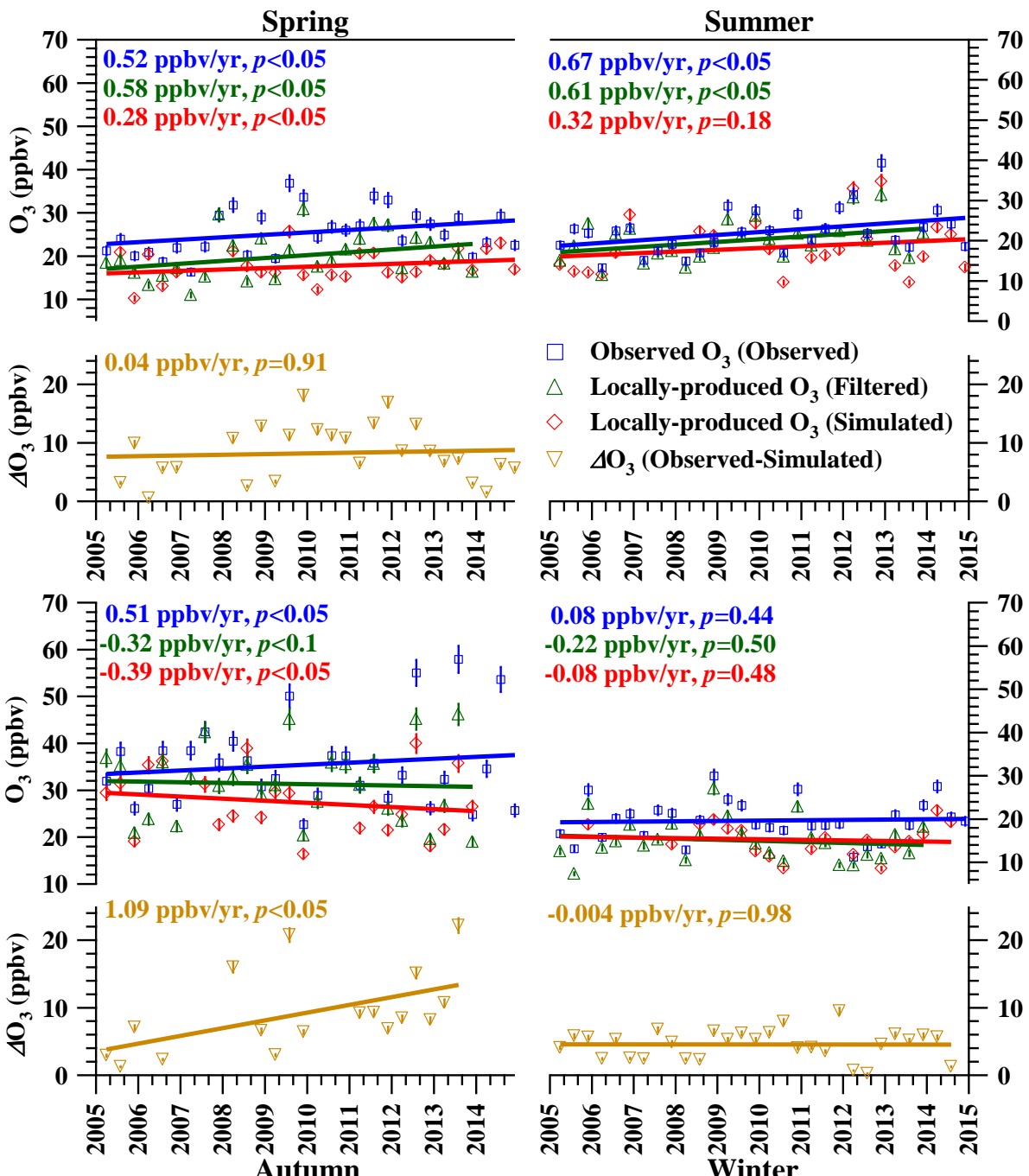

**Figure 4.** Trends of locally-produced $O_3$ simulated by OBM (red line), observed $O_3$ (blue line: overall observed $O_3$; green line: locally-produced $O_3$ (filtered), *i.e.*, observed $O_3$ with hourly wind speed <2 m/s), and regional $O_3$ (gold line, $\Delta O_3 = O_3$ overall observed $- O_3$ simulated) in four seasons at TC during 2005–2014. Note: all the data are based on daytime hours (0700-1900 LT). The regional $O_3$ in summer was negligible and is not shown in the graph. Error bars represent 95% confidence interval of the averages.

To further investigate the regional transport from the PRD region to Hong Kong, the observed $O_3$ values at PRD sites and at TC site in four seasons during 2006–2014 are compared (Figure 5). Generally, the observed $O_3$ levels in PRD were all higher than those at TC in four seasons ($p<0.05$). Considering that the PRD is upwind of Hong Kong in spring/autumn/winter (Ou *et al*.

5  2015), high $O_3$-laden air in the PRD region could transport to Hong Kong in the three seasons. Moreover, comparable long-term trends were found between the sites in PRD and the TC site in spring (PRD: 0.49±0.06 ppbv yr$^{-1}$; TC: 0.51±0.05 ppbv yr$^{-1}$) and winter (PRD: 0.30±0.06 ppbv yr$^{-1}$; TC: 0.23±0.05 ppbv yr$^{-1}$), indicating that regional transport in spring and winter was stable in these years. In comparison, autumn $O_3$ level in inland PRD increased (0.84±0.08 ppbv yr$^{-1}$) more rapidly than in Hong Kong (0.67±0.07 ppbv yr$^{-1}$), implying an elevated regional contribution to Hong Kong. Therefore, the differences

10  of observed $O_3$ between PRD and TC in spring/autumn/winter were consistent with the above calculations of average $\Delta O_3$, confirming regional contribution to the observed $O_3$ in Hong Kong. In contrast, though the increasing rate of $O_3$ level in PRD was much faster than at TC in summer, the impact of regional transport from PRD region was insignificant due to the dominance of southerly and southeasterly winds from the South China Sea.

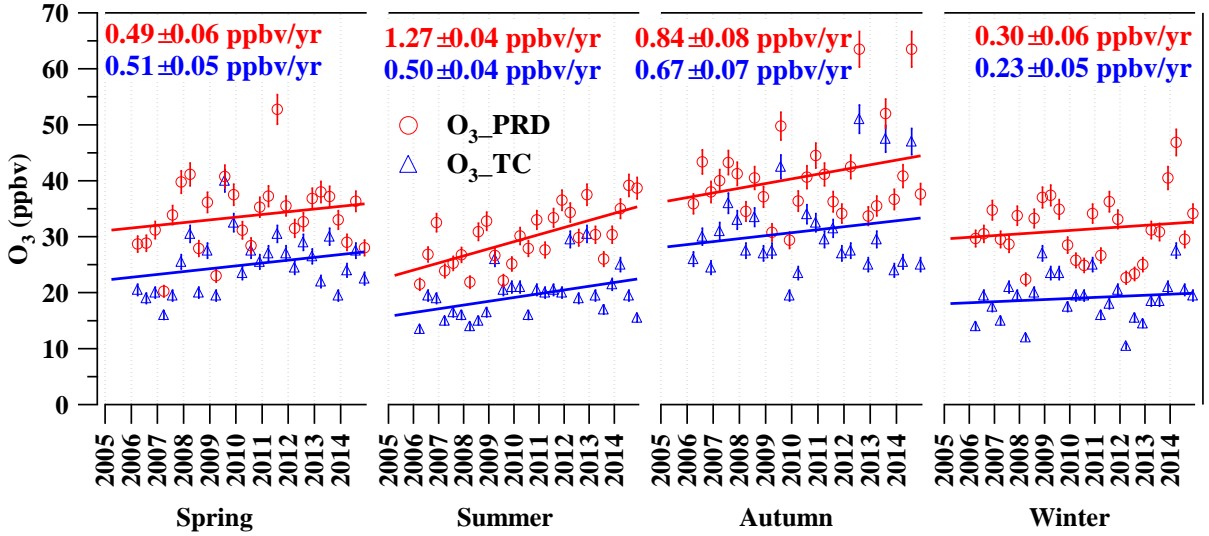

**Figure 5.** Trends of observed $O_3$ in inland PRD (red line) and at TC (blue line) in four seasons during 2006–2014. Each data point in the figure is obtained by averaging hourly values into a monthly value. Note: Due to the location of sites which might have $O_3$ transport to TC (Guo *et al*., 2009), three regional background sites (*i.e.*, Wanqingsha, Jinguowan and Tianhu) and an urban site (*i.e.*, Haogang) in Dongguan are used for comparison. Error bars represent 95% confidence interval of monthly averages.

Overall, locally-produced $O_3$ (simulated) in Hong Kong varied by season, showing an increase in spring, a decrease in autumn and no change in summer and winter. The elevated observed $O_3$ in spring/summer/autumn was mainly attributed to the increase of locally-produced springtime $O_3$ and constant regional contribution, increased summertime in-situ photochemical reactivity, and regional contribution in autumn. Moreover, since the $NO_x$ and NO levels significantly decreased during the last decade (Figure S8), the reduced effect of NO titration, to a certain extent, made contribution to local $O_3$ levels. The effect of NO titration has also been reported in other areas (*i.e.*, Beijing, Taiwan, Guangdong Province in China and Osaka in Japan) (Chou *et al.*, 2006; Itano *et al.*, 2007; Tang *et al.*, 2009; Li *et al.*, 2014). To confirm the reduction of NO titration in this study, the variation of $O_x$, the total oxidant estimated by $O_3+NO_2$ was investigated. According to the reaction of NO titration ($NO+O_3\rightarrow NO_2+O_2$), the sum of $O_3$ and $NO_2$ (*i.e.*, total oxidant $O_x$) remained essentially constant regardless the variation of NO (Chou *et al.*, 2006). Indeed, the mixing ratio of local $O_x$ (filtered by wind speed $< 2m/s$) showed no significant change ($p=0.42$) at TC site, during 2005-2013 (Figure 6), suggesting that the increase of $O_3$ was a result of the reduced NO titration. The reduction of NO titration was also confirmed by the increasing $NO_2/NO_x$ ratio at a roadside site (Mong Kok) in Hong Kong. The $NO_2/NO_x$ emission ratio is a parameter that can be used to examine the variation of NO titration (Carslaw, 2005; Yao *et al.*, 2005; Dallmann *et al.*, 2011; Tian *et al.*, 2011; Ning *et al.*, 2012; Lau *et al.*, 2015). Generally, higher ratios of $NO_2/NO_x$ mean a lower potential of $O_3$ titration by NO, resulting in higher $O_3$. Indeed, the $NO_2/NO_x$ ratio at Mong Kok significantly increased, with an enhanced traffic related $NO_2/NO_x$ ratios observed at night, from 2005 to 2014 ($p<0.01$), leading to increased local $O_3$ levels (Figure 7). This finding was supported by the Hong Kong emission inventory, which indicated that the $NO_x$ emission decreased from 1997 to 2014 in Hong Kong (HKEPD, 2016), and studies conducted by Tian *et al.* (2011) and Lau *et al.* (2015), who found an increasing trend of primary $NO_2$ emission in Hong Kong due to several diesel retrofit programs in 1998-2008.

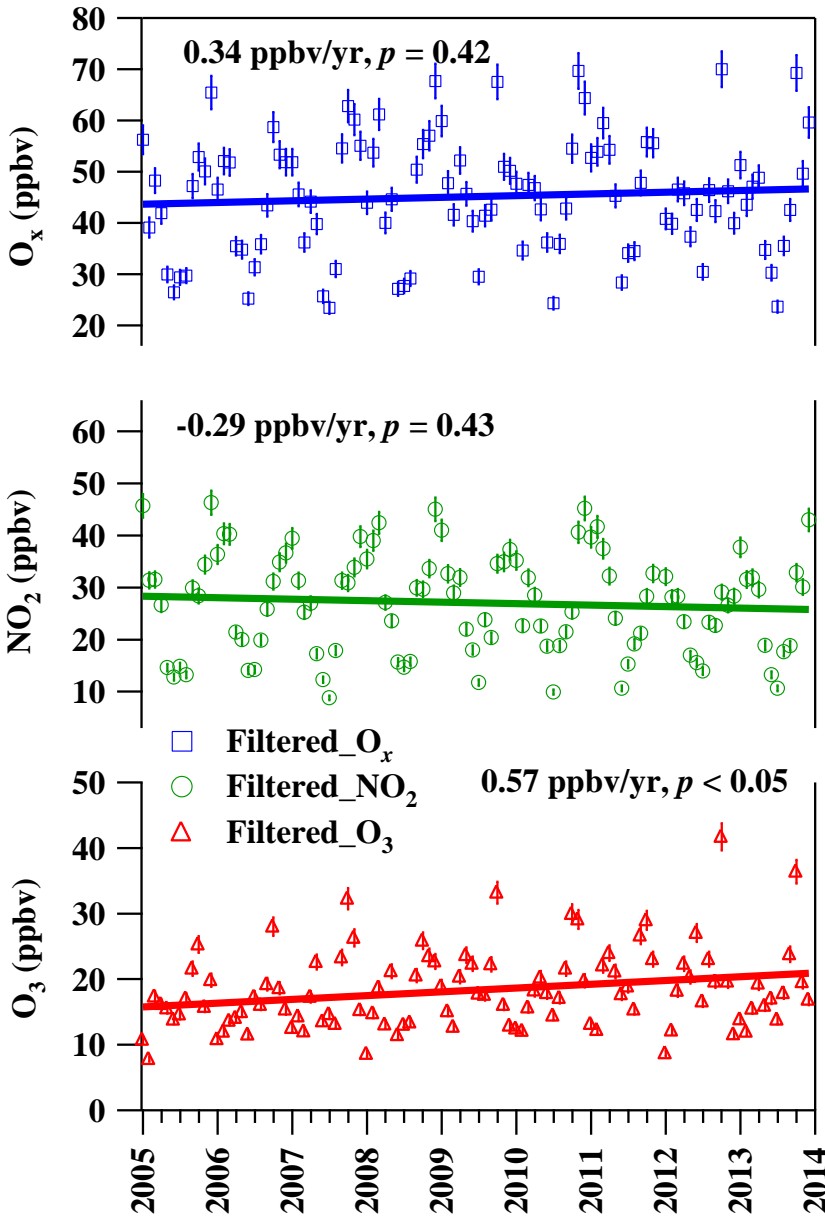

**Figure 6.** Annual trend of $O_x$, $O_3$ and $NO_2$ (filtered) at TC in 2005–2013. Error bars represent the 95% confidence interval of the averages.

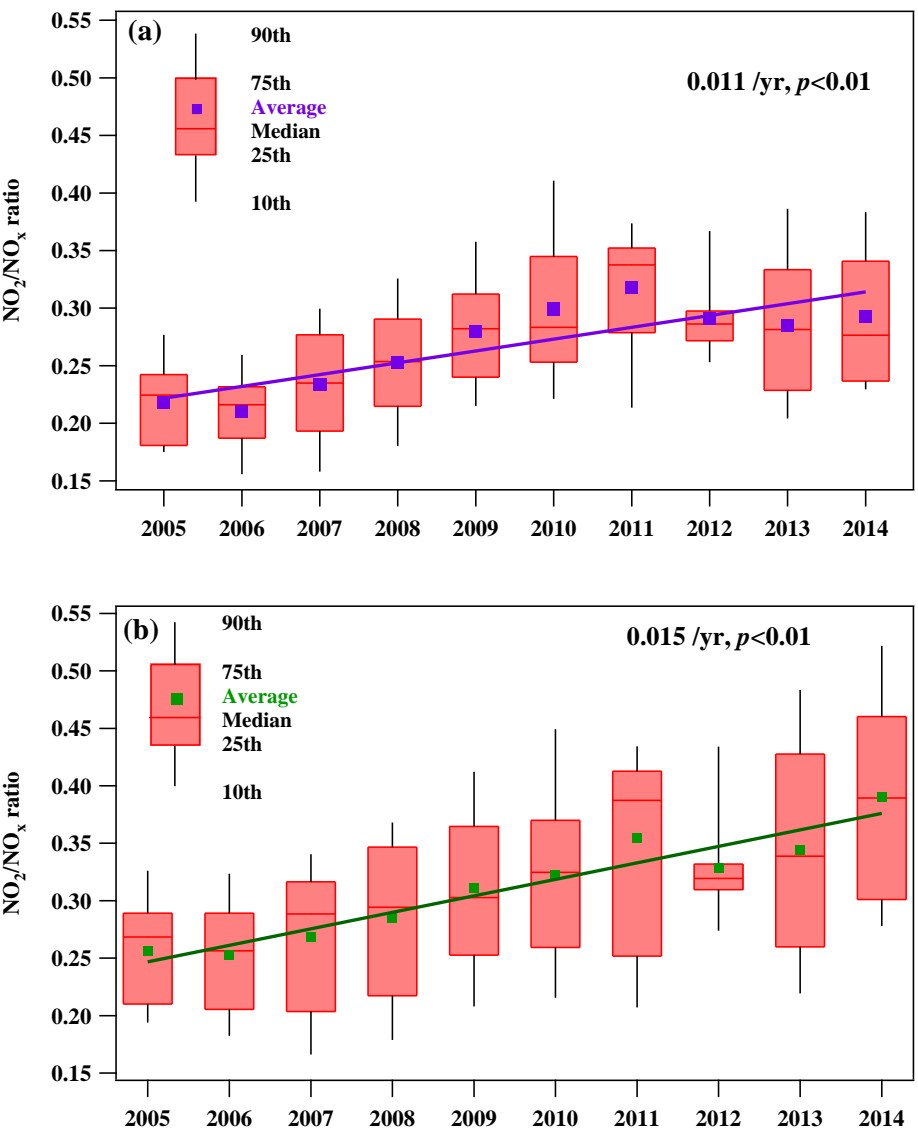

**Figure 7.** Annual trend of monthly average $NO_2/NO_x$ ratio at MK site in daytime **(a)** and night-time **(b)** in Hong Kong in 2005–2014. Hourly observations of $NO_2$ and $NO_x$ are obtained at MK from the HKEPD (http://epic.epd.gov.hk/ca/uid/airdata). Note that data of October in 2014 are excluded due to the impact of Occupy Central event in Hong Kong.

Apart from the regional and local impact on $O_3$ trends, the impact of variations of baseline $O_3$ was also considered. Oltmans *et al*. (2013) reported that $O_3$ at mid-latitudes of the Northern Hemisphere was flat or declining during 1996-2010 and the limited data in the subtropical Pacific suggested very little change during the same period. Thus, the $O_3$ trend in Hong Kong might be

unaffected or underestimated given the flat or decline of baseline $O_3$. Therefore, the increasing trend of $O_3$ in Hong Kong over the last decade was the integrated influence of its precursors, meteorological parameters and regional transport.

### 3.3 Ozone - precursor relationships

Ozone - precursor relationships are critical to determine the reduction plan of precursors for future $O_3$ control. In this study, the
RIR of major $O_3$ precursors was calculated by the OBM, to directly reflect the $O_3$ alteration in response to the percentage changes of its precursors (see Section 2.3). Furthermore, the long-term trend of RIR was used to evaluate the variation of the sensitivity of $O_3$ formation to each individual precursor. Figure 8(a) shows the average RIR values of $O_3$ precursors in the four seasons during the last decade. The RIR values of TVOCs (AVOC+BVOC, where AVOC = anthropogenic VOC and BVOC = biogenic VOC, see Section 3.4 for the definition of BVOC) ranged from $0.78 \pm 0.04$ to $0.89 \pm 0.04$, followed by CO ($0.32 \pm 0.01$
to $0.37 \pm 0.01$) in all seasons, suggesting the dominant role of TVOCs in photochemical $O_3$ formation. Among TVOCs, AVOCs had their highest RIR value in winter ($0.80 \pm 0.03$, $p<0.05$) and lowest in summer ($0.39 \pm 0.02$, $p<0.05$). Since RIR values are highly dependent on precursor mixing ratios (Eq. 5), the difference of RIR values of AVOCs in the four seasons was mainly caused by seasonal variations of observed AVOC levels (Figure 3). In contrast, BVOCs had the highest RIR in summer ($0.47 \pm 0.02$, $p<0.05$), followed by autumn, spring and winter ($0.30 \pm 0.02$, $0.28 \pm 0.02$ and $0.09 \pm 0.01$, respectively). The higher
RIR of BVOCs in summer was mainly due to the higher biogenic emissions in summer. In addition, higher photochemical reactivity of BVOCs also contributed to higher RIR of BVOCs (Atkinson and Arey, 2003; Tsui *et al.*, 2009). The RIR values of $NO_x$, in contrast, were negative in all seasons, indicating that reducing $NO_x$ would lead to an increase of photochemical $O_3$ formation. The RIR values of $NO_x$ were lower in spring ($-1.15 \pm 0.02$) and summer ($-1.22 \pm 0.02$) than in autumn ($-1.05 \pm 0.02$) and winter ($-1.00 \pm 0.01$, $p<0.05$), suggesting that reducing $NO_x$ would increase more $O_3$ in spring and summer. The
aforementioned findings were consistent with the results of previous studies conducted in autumn in Hong Kong, which were based on modeled and observed VOC/$NO_x$ ratios (Zhang *et al.*, 2007; Cheng *et al.*, 2010; Ling *et al.*, 2013; Guo *et al.*, 2013). The relationship analyses suggests that the $O_3$ formation in Hong Kong was VOC-limited in all seasons in these years; that is, the $O_3$ formation was dominated by AVOCs in winter and was sensitive to both AVOCs and BVOCs in the other three seasons, whereas reducing $NO_x$ emissions enhanced $O_3$ formation, more so in spring and summer. The findings suggest that
simultaneous cut of AVOCs and $NO_x$ (which is often the case in real situation) would be most effective in $O_3$ pollution control in winter, but least efficient in summer.

Figure 8(b) presents the long-term trends of RIR values of $O_3$ precursors from 2005 to 2014. The RIR values of TVOCs and $NO_x$ increased at an average rate of $0.014 \pm 0.012$ yr$^{-1}$ ($p<0.05$) and $0.009 \pm 0.01$ yr$^{-1}$ ($p<0.05$), respectively, while the RIR of CO decreased at an average rate of $-0.014 \pm 0.007$ yr$^{-1}$ ($p<0.01$). The evolution of RIR values suggested that the $O_3$ formation was
more sensitive to TVOCs and less to CO and $NO_x$, indicating that VOCs reduction strategies would be more effective on $O_3$ control. The decreasing sensitivities of both CO and $NO_x$ to $O_3$ formation were consistent with the decrease of their mixing ratios during the last decade. The sensitivity of $O_3$ formation to TVOCs increased although the 10-year levels of TVOCs showed no significant trend, which might be attributed to the variations of speciated VOC levels and the VOC/$NO_x$ ratios in

these years (Ou *et al.*, 2015). Furthermore, the monthly variation of VOC/NO$_x$ ratios showed a significant decreasing trend at a rate of -0.02 yr$^{-1}$ ($p<0.05$) (see Figure S17), indicating that VOC reduction became more effective in reducing O$_3$ in the past 10 years, which is consistent with the conclusions from the above modeling results.

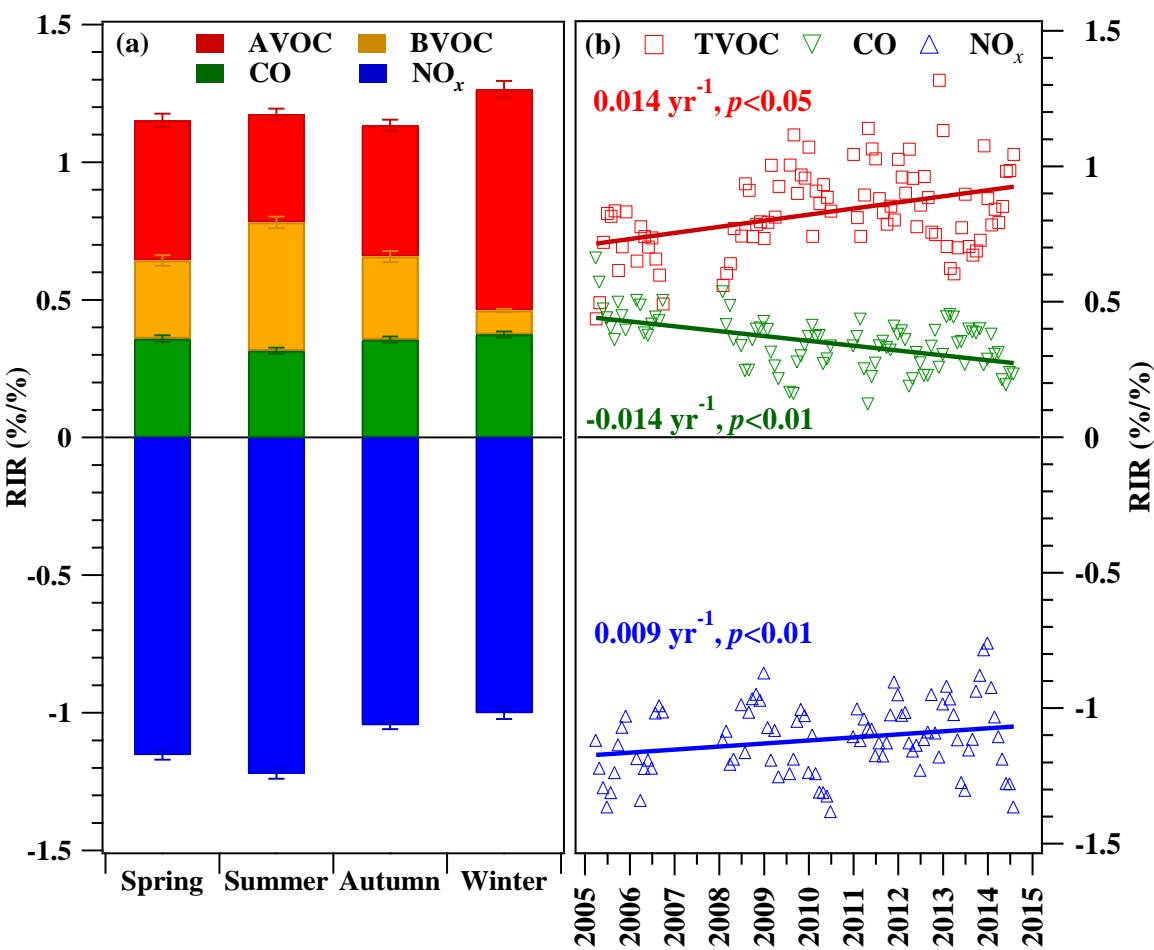

**Figure 8.** (a) Average RIR values of O$_3$ precursors in the four seasons; and (b) Trends of RIR values of O$_3$ precursors at TC from 2005 to 2014. AVOC represents anthropogenic VOCs, including 20 VOC species. BVOC means biogenic VOCs, including isoprene. Note: all the data are based on daytime hours (0700-1900 LT).

**3.4 Contribution of VOC groups to O$_3$ formation**

Since the local O$_3$ production was VOC-limited in Hong Kong, it is important to study the contribution of VOC species to the O$_3$ formation. To facilitate analysis and interpretation, AVOC species were categorized into three groups, namely, AVOC (Aromatics) (including benzene, toluene, *m/o*-xylenes, ethylbenzene and three trimethylbenzene isomers), AVOC (Alkenes) (including propene, three butene isomers and 1, 3-butadiene) and AVOC (Alkanes) (including propane, *n/i*-butanes,

$n/i$-pentanes, $n/i$-hexanes and $n$-heptane). It is noteworthy that $C_2$ hydrocarbons were not included in the groups due to high missing rates of the $C_2$ data The variations of the daytime averaged contributions of four VOC groups (i.e., AVOC (Alkanes/Alkenes/Aromatics) and BVOC) to $O_3$ mixing ratios were calculated by OBM and shown in Figure 9. Two scenarios were selected for data analysis. The first scenario was "origin", which used all originally measured data as input. The second scenario was "AVOC (Aromatics/Alkenes/Alkanes) or BVOC group", which excluded each of the four VOC groups in turn from the input data in the "origin" scenario. Hence, the contributions of VOC groups ((AVOC (Aromatics/Alkenes/Alkanes) and BVOC) were obtained from the difference of simulated $O_3$ between the scenario "origin" and the related "VOCs group (AVOC (Aromatics/Alkenes/Alkanes) or BVOC))". Clearly, the contribution of AVOC (Aromatics) to $O_3$ mixing ratios significantly decreased with an average rate of -0.23±0.01 ppbv yr$^{-1}$ ($p<0.05$), while AVOC (Alkenes) made increasing contribution to $O_3$ mixing ratio ($p<0.05$), and BVOC and AVOC (Alkanes) showed no significant changes ($p>0.05$). The decreased contribution of AVOC (Aromatics) to the $O_3$ mixing ratio was likely due to the decrease of $C_6$-$C_8$ aromatics, consistent with previous studies which found that aromatic levels decreased during 2005-2013 in Hong Kong (Ou $et$ $al$., 2015). In fact, the Hong Kong Government has implemented a series of VOC-control measures since 2007 (HKEPD, 2016). From April 2007, the Air Pollution Control (VOCs) Regulation was implemented to control VOC emissions from regulated products, including architectural paints/coatings, printing inks and six selected categories of consumer products. In January 2010, the regulation was extended to control other high VOC-containing products, namely vehicle refinishing paints/coatings, vessel and pleasure craft paints/coatings, adhesives and sealants. The reduced contribution of AVOC (Aromatics) to $O_3$ formation in these years also agreed well with the decreasing $O_3$ production rate of aromatics in autumn at TC from 2002 to 2013 reported by Xue $et$ $al$. (2014). Furthermore, source apportionment results from Lyu $et$ $al$. (2017) showed that solvent related VOCs decreased at a rate of 204.7±39.7 pptv yr$^{-1}$ at TC site, confirming that the reduction of solvent usage in these years was effective in decreasing the contribution of aromatics to the $O_3$ production.

In contrast, the contributions of AVOC (Alkenes) to $O_3$ production in these years showed a significant increasing trend with a rate of 0.14±0.01 ppbv yr$^{-1}$ ($p<0.05$), perhaps attributed to the increased emissions of alkenes from traffic related sources. During 2005-2014, the Hong Kong government launched a series of measures to reduce vehicular emissions, including diesel, LPG and gasoline vehicles (http://www.epd.gov.hk/epd/english/environmentinhk/air_/prob_solutions/air_problems.html). Among the measures, the diesel commercial vehicle (DCV) program (2007-2013) was shown to be effective in reducing the emission of alkenes from diesel vehicles (Lyu $et$ $al$., 2017). However, gasoline and LPG vehicular emissions caused ambient alkenes to increase during the same period due to the increasing of number of LPG/gasoline vehicles and some short-term/non-mandatory measures (Lyu $et$ $al$., 2017). In consequence, the overall emissions of alkenes from traffic related sources increased during 2005-2014, leading to the increased contribution of AVOC (alkenes) to $O_3$ formation (Lyu $et$ $al$., 2017).

Unlike AVOC (Aromatics/Alkanes), the contribution of AVOC (Alkanes) to $O_3$ formation during 2005-2014 showed no significant change ($p=0.23$), perhaps due to their low photochemical reactivities (see Table S10) despite the increase of total alkane levels in the atmosphere in 2005-2013 (Ou $et$ $al$., 2015), and the decrease of LPG-related alkanes in 2013-2014 (Lyu $et$

*al.*, 2016). In addition, the seasonal variation of O₃ formation, of which the reaction rates of alkanes with OH radicals were high in winter, would also blur the trend.

Furthermore, BVOC showed no evident change in the contribution to O₃ mixing ratios during the last decade (*p*=0.57), which was likely attributed to the no significant change of isoprene levels at TC site during 2005-2014 (shown in Figure S18). In this study, isoprene was defined as biogenic VOCs. The main known sources of isoprene are biogenic and anthropogenic (Borbon *et al.* 2001; Barletta *et al.*, 2002; Reiman *et al.*, 2000). It is noteworthy that according to the tunnel study in Hong Kong (Ho *et al.*, 2009), vehicular emissions of isoprene are not significant in this city. Another tunnel study in the PRD region (Tsai *et al.*, 2006) found that isoprene was not present in diesel-fueled vehicular emissions in Hong Kong, likely related to variations of fuel types and vehicular engines used in different countries (Ho *et al.*, 2009). In addition, in low latitude areas like Hong Kong, with a high level of plant coverage (more than 70%), isoprene is mainly produced by biogenic emissions. The source of isoprene at TC site has been also investigated and confirmed by previous long-term source appointment studies, which reported that during 2005-2013 about 90% of isoprene was emitted from biogenic emissions and the contribution of traffic sources was not significant (<5%) (Ou *et al.*, 2015). Therefore, in this study the traffic sources of isoprene in Hong Kong was disregarded and isoprene was defined as biogenic VOCs.

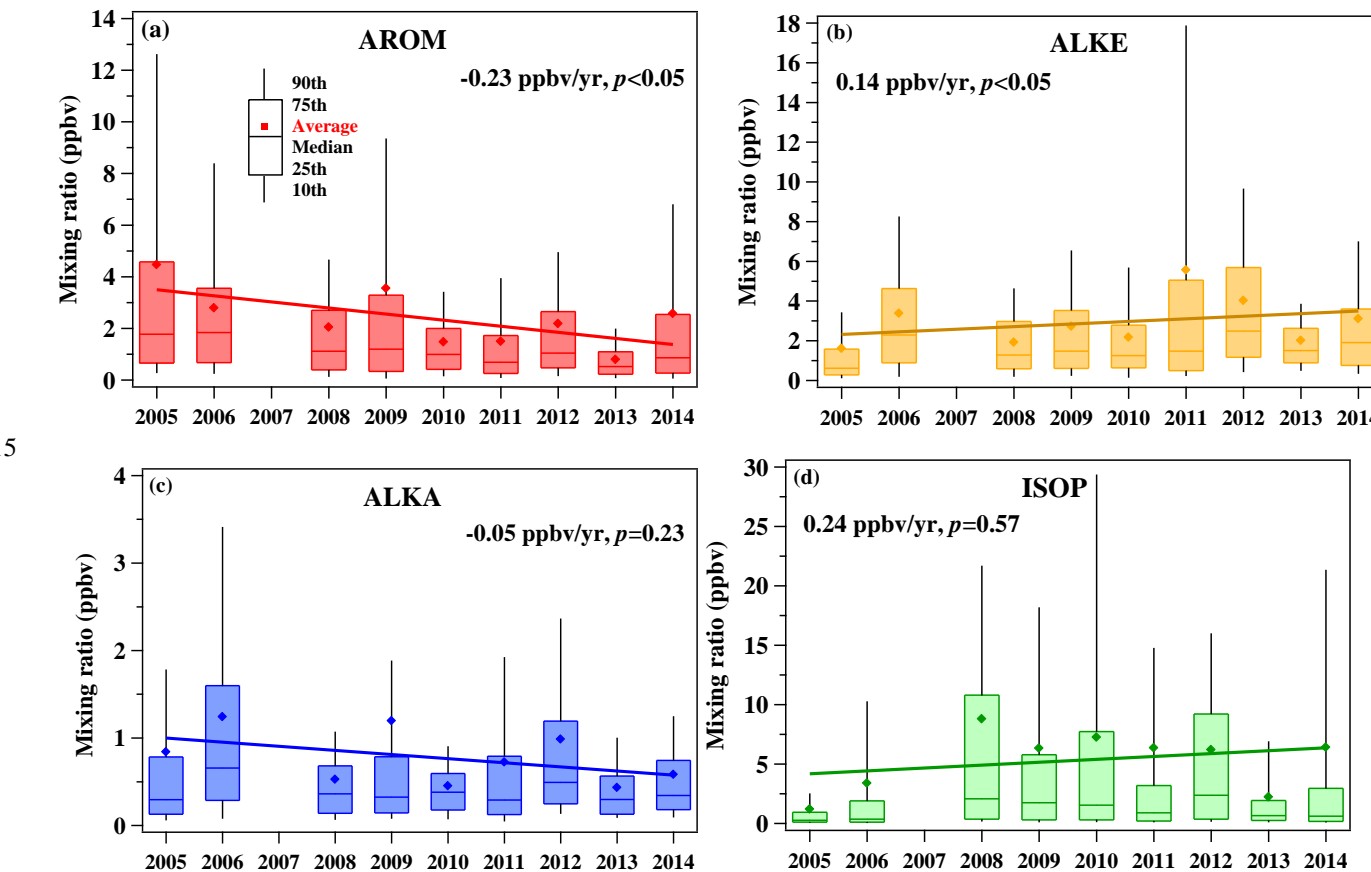

**Figure 9.** Trends of the daytime averaged contribution of four VOC groups to $O_3$ mixing ratio: (a) AVOC (Aromatics), (b) AVOC (Alkenes), (c) AVOC (Alkanes) and (d) BVOC at TC during 2005–2014.

## 4 Conclusions

In this study, the long-term trends of $O_3$ and its precursors ($NO_x$, TVOCs and CO) were analyzed at TC from 2005 to 2014. It was found that $NO_x$ and CO decreased while TVOCs remained unchanged, suggesting the effective reduction of some emissions in Hong Kong. However, ambient $O_3$ levels increased in these years and the locally-produced $O_3$ showed different variations in the four seasons, reflecting the complexity of photochemical pollution in Hong Kong. To effectively control locally-produced $O_3$, VOC control plays a vital role, since $O_3$ formation in Hong Kong was shown to be VOC-limited in these years. Moreover, trend studies found that the sensitivity of $O_3$ formation gradually increased with VOCs and decreased with $NO_x$ and CO, indicating that controlling VOCs will be increasingly effective for $O_3$ control in the future. Among the VOCs, the contribution of aromatics to $O_3$ formation decreased from 2005-2014, consistent with their declining abundance over this period and implying effective control measures of solvent-related sources. By contrast, the contribution of anthropogenic alkenes increased, suggesting a continuing need for the control of traffic-related sources. In addition, of the four seasons, the highest AVOC sensitivity and relatively low $NO_x$ sensitivity to $O_3$ formation concurrently appeared in winter, suggesting that winter is the best time for $O_3$ control. Lastly, in addition to locally-produced $O_3$, regional transport of $O_3$ from the PRD region made a substantial contribution to ambient $O_3$ in Hong Kong and even increased in autumn. In the future, the Hong Kong government should collaborate closely with Guangdong province to mitigate $O_3$ pollution in this region.

*Acknowledgments.*

We thank Hong Kong Environmental Protection Department for providing us the data. This work was supported by the Natural Science Foundation of China (41275122), the Research Grants Council (RGC) of the Hong Kong Government of Special Administrative Region (PolyU5154/13E, PolyU152052/14E, PolyU152052/16E and CRF/C5004-15E), the Innovation and Technology Commission of the HKSAR to the Hong Kong Branch of National Rail Transit Electrification and Automation Engineering Technology Research Center, and the Hong Kong Polytechnic University PhD scholarships (project #RTTA). This study is partly supported by the Hong Kong PolyU internal grant (1-BBW4 and 1-BBYD).

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
