# Peer review of "Long term O3-precursor relationships in Hong Kong: Field observation and model simulation"

_Atmospheric Chemistry and Physics, 2017_

## Referee Comment (RC1)

**General comment**

A valuable long term dataset is described as a basis for an interesting long term trend study on ozone and its precursors, which can be used to track how emission changes and some abatement strategies are effecting air quality in Hong Kong, and how best to deal with air quality issues into the future. However, much more detail on the scientific approaches and applied methods (measurement techniques, modelling and data analysis/filtering) are needed as detailed below. Once these issues have been adequately addressed, this study can be accepted for publication in ACP.

**Specific comments**

More details of the measurement techniques, in particularly the QA/QC methodologies and approaches for the GC analysis is needed (stating computer programme used for autocalibration is not good enough). I find a detection limit of 2 pptv using this technique quite low, especially for an hourly measurement.

Different speciated sets of VOCs have been targeted in previous studies, such as 30 C2 –C10 in previous studies at MK and HT (Lyu et al., 2016). Why have you focused on a different speciation of 21 species in the present study? Given the extensive annual VOC information available at other sites (local and regional) it would be useful to compare seasonally averaged measurements at as many different sites as possible (in a Table?).

Difficult to separate the background and regional effects on observed ozone formation from the local chemical production and more detail into how the authors have done this is needed. What are the sensitivities in the model to these (potentially high) uncertainties in initial concentrations of ozone precursor?.

Set the overall air quality picture in HK in perspective - How do the observed precursor trends and ratios (e.g. NO2/NOx) compare to other high population industrial urban areas of the world on a year by year seasonal basis? There is a wide range of comparable data in the literature.

Difficult to relate specific plots with specific trends discussed in the main text.

Confusing in parts and difficult to read/follow.

Missing reactivity – This study is linked to the measurement of 21 VOC precursors, which may be the main source of ozone in the region in the inventories (show this?). However, how much primary reactivity is potentially missing (e.g. long chain/branched alkanes and alkenes?), on a year by year basis? Some discussion on the missing reactivity and how this may affect the presented results is warranted.

**Technical comments**

Fig 1 – "cities" of regions? A more detailed map/maps of the local and regional environs including distance to downtown HK would be useful. What are the prevailing wind directions in each season?

Page 5 (modelling) - Reference for TUV photolysis modelling needed – model sensitivity to cloud and haze days? Loss of radicals to aerosol sensitivity studies? MCM is not an "explicit" mechanism. It is described as "semi-explicit" – more complex and complete chemistry than CB05.

Why was a simple BLH/deposition sensitivity run not performed?- sensitivity?

Did the model runs include a "spin up" time to get the radical intermediates into steady state? i.e. run for 2-4 consecutive days and then taking the data for the last 24 hours? This could significantly effect the model O3 coming from the model secondary chemistry and the RIR calculations if the intermediates are not in steady state.

The reasoning behind the removal of "background" and "local" concentrations to give an "observed" value are vague and questionable. Their influence will be highly variable with respect to the meteorology and time of the year. What are these concentrations and how were they derived (by wind direction? hourly, monthly, seasonally?). Again, I would like to see the seasonal concentrations of VOC measurements at other regional and local sites compared and contrasted to the current data set in some detail and how sensitivity to this calculation effects the deconvolution of local photochemical ozone formation. The authors state that the "NO2 emitted from the inland PRD is slightly more likely (?) to arrive at TC in winter and spring

than in summer and fall, the differences in travel time among the seasons are relatively small and it is difficult to be precise with seasonal average estimates of NO2 lifetime and travel time".

Therefore, the errors and limitations in understanding the effect of regional/background NO2 on the observed O3 are high here (which the authors acknowledge) – what are the uncertainties associated with this and what is the sensitivity to the model runs?

Eq 7 – delta kro2+no[RO2][NO]?

Fig 3. Not sure how significantly different the inter-annual trends of the TVOC in spring and autumn are?

Fig 4. Why are the locally produced simulated and filtered trends so different in most seasons?

Page 10 – "The different inter-annual trends of NOX and TVOCs in spring/autumn from those in summer/winter were probably because marine air significantly diluted air pollution in summer while continental air masses remarkably burdened air pollution in winter, which concealed the decreased local emissions of NOX and TVOCs in summer and winter (Wang et al., 2009)" – do you have winder sector data to show evidence for this?

Define and show NO titration reaction as a separate Equation.

Further investigation found that temperature and solar radiation in summer indeed increased in these years (p<0.05)" Show this data.

Page 11 – Section 3.2 repeats itself somewhat

Page 12 – the MCM is not "explicit". Should also be (p<0.05)?

"locally-produced  $O_3$  (filtered) values clearly showed similar trends to locally-produced  $O_3$  (simulated) in spring, autumn and winter (p=0.07, 0.09 and 0.93, respectively)" - where is the plot showing these trends?

"Unlike in spring, though the observed and locally-produced O3 (filtered) displayed increasing trends in summer (0.70 $\pm$ 0.34 ppbv/yr and 0.66 $\pm$ 0.41 ppbv/yr, respectively; p<0.05)," – these

trends are not the same as shown in Fig 4?

Page 13 – "the total solar radiation  $(0.24\pm0.16 \text{ MJ}\cdot\text{m}^{-2}\text{yr}^{-1}, \text{ p}<0.01)$  and temperature  $(0.095\pm0.034 \text{ }^{0}\text{C/yr}, \text{p}<0.05)$  in summer significantly increased during the past 10 years" – odd units! Solar radiation should be given as irradiance in W m-2. Temperature in K. Be consistent as to how you write units – K/y or K y-1. Why has the solar radiation increased in the summer over 10 years? Less haze?

Fig 9. Why is the data not presented as monthly averages or even by season – which would be clearer and possibly give more information?

Page 20 (contribution of VOC groups) – I would have liked to see the year by year seasonal trends of the individual TVOC groups plotted – i.e. aromatics, alkanes, alkenes and BVOC. This would give a more comprehensive overview of the TVOC trend and how it evolved/is evolving.

Page 21. Basic detail of the HK government emissions reduction plan need to be outlined (with the reference website placed in the References section).

"photolysis rates of alkanes" - I am sure alkanes do not photolyse in the atmosphere!

---

## Author Response (AR1)

**Response to Referee 1's comments**

**General comment**

*A valuable long term dataset is described as a basis for an interesting long term trend study on ozone and its precursors, which can be used to track how emission changes and some abatement strategies are effecting air quality in Hong Kong, and how best to deal with air quality issues into the future. However, much more detail on the scientific approaches and applied methods (measurement techniques, modelling and data analysis/filtering) are needed as detailed below. Once these issues have been adequately addressed, this study can be accepted for publication in ACP.*

We highly appreciate the reviewer for the positive comments and constructive suggestions. In general, all these comments have been addressed properly with addition of required words, figures and tables in the revised manuscript, particularly with great details of the methods. Furthermore, a number of modelling sensitivity tests have been conducted and the new results have been added to support our conclusions. Also, many sentences have been reorganised to either reduce confusing or improve readability. In the following section, the author's responses (in blue) are immediately after the reviewer's comments (in black), with the changes in manuscript at the end (in italic).

**Specific comments**

**Comment 1.1**

More details of the measurement techniques, in particularly the QA/QC methodologies and approaches for the GC analysis is needed (stating computer programme used for auto calibration is not good enough). I find a detection limit of 2 pptv using this technique quite low, especially for an hourly measurement.

**Response:**

More detailed description about the measurement techniques, such as instrumentation, detection limit, resolution, and QA/QC, has been added in the revised manuscript. In addition, it is not surprising to achieve a detection limit of 2 pptv for using this technique. In many previous studies (not only in Hong Kong), a comparable low detection limit was reported by using the same measurement techniques, such as Xie *et al.* (2008) and Yuan *et al.* (2009).

*References:*

Xie, X., *et al*. (2008), "Estimation of initial isoprene contribution to ozone formation potential in Beijing, China", Atmos. Environ., 42, 6000–6010, doi:10.1016/j.atmosenv. 2008.03.035.

Yuan, Z., *et al*. (2009). "Source analysis of volatile organic compounds by positive matrix factorization in urban and rural

5   environments in Beijing." Journal of Geophysical Research Atmospheres 114(16).

*Revision in the manuscript:*

*Page 4, Line 6:*

*"2.2 Measurement techniques*

10   *Hourly observations of $O_3$, CO, $SO_2$, $NO$-$NO_2$-$NO_x$ and meteorological parameters at TC from 2005 to 2014 were obtained from the HKEPD (http://epic.epd.gov.hk/ca/uid/airdata). Briefly, $O_3$ was measured using a commercial UV photometric instrument (Advanced Pollution Instrumentation (API), Model 400A) with a detection limit of 0.6 ppbv. CO was measured with a gas filter correlation CO analyser (Thermo ElectronCorp. (TECO), Model 48C) with a detection limit of 0.04 ppm. $SO_2$ was measured using a pulsed fluorescence analyser (TECO, Model 43A) with a detection limit of 1.0 ppbv. $NO$-$NO_2$-*

15   *$NO_x$ were detected using a commercial chemiluminescence with an internal molybdenum converter (API, Model 200A) and a detection limit of 0.4 ppbv. All the time resolutions for these gas analysers were 1 hour. To ensure a high degree of accuracy and precision, the QA/QC procedures for gaseous pollutants were identical to those in the US air quality monitoring program (http://epic.epd.gov.hk/ca/uid/airdata). The accuracy of the monitoring network was assessed by performance audits, while the precision, a measure of the repeatability, of the measurements was checked in accordance with HKEPD's*

20   *quality manuals. For the gaseous pollutants, the accuracy and precision within the limits of ±15 and ±20 % were adopted, respectively (HKEPD 2015).*

*Real-time VOC data at TC were also measured by the HKEPD. An online GC-FID analyser (Synspec GC 955, Series 600/800) was used to collect VOC speciation data continuously with a time resolution of 30 minutes. The VOC analyser consists of two separate systems for detection of $C_2$–$C_5$ and $C_6$–$C_{10}$ hydrocarbons, respectively. Detailed description about*

25   *the real-time VOC analyser can be found in Lyu et al. (2016). Twenty-eight $C_3$-$C_{10}$ VOC species were identified and quantified using this method. In terms of the QA/QC for VOC analysis, built-in computerized programs of quality control systems such as auto-linearization and auto-calibration were used. Weekly calibrations were conducted by using NPL standard gas (National Physical Laboratory, Teddington, Middlesex, UK). In general, the detection limits of the target VOCs ranged from 2 to 56 pptv. The accuracy of each species measured by online GC-FID was determined by the*

30   *percentage difference between measured mixing ratio and actual mixing ratio based on weekly span checks and monthly calibrations. The precision was based on the 95% probability limits for the integrated precision check results. The accuracy*

*of the measurements was about 1-7%, depending on the species, and the measurement precision was about 1-10% (Table S1). In addition, the quality of the real-time data was assured by regular comparison with whole-air canister samples collected and analysed by University of California at Irvine (UCI). More details can be found from previous studies in Hong Kong (Xue et al., 2014; Ou et al., 2015; Lyu et al., 2016).*

5       *For data analysis, linear regression and error bars represented as 95% confidence intervals were used. Trends of $O_3$ and its precursors with a p value < 0.05 were considered significant (Guo et al., 2009)."*

**Comment 1.2**

Different speciated sets of VOCs have been targeted in previous studies, such as 30 $C_2$ –$C_{10}$ in previous studies at MK and

HT (Lyu *et al*., 2016). Why have you focused on a different speciation of 21 species in the present study?

10   **Response:**

      The main reason why not all detected species were included in the present study is because of high percentage of missing data for several species at TC site over 2005-2014. This problem was very serious in some years and also existed for other sites, such as MK and HT, over the years. As this study looks into the trend of $O_3$, those species with high missing data were excluded to eliminate the influence caused by different input of VOC species.

15       We admitted that the limited number of species would underestimate the $O_3$ production. To assess this impact, the missing reactivity has been examined and the results are presented in the Response to Comment 1.9.

      In addition, *m*-xylene listed in Tables S1 and S2 actually represents *m/p*-xylene (*i.e.*, *m*-xylene and *p*-xylene) as the VOC analysis method used in this study could not divide the two species apart. To reflect this fact, the number of VOC species adopted in the study has been corrected from 21 to 22. It is noteworthy that this change would not affect our results and

20   conclusion, because the three xylene isomers have the same CB05 model specie (*i.e.*, XYL, see Table S2).

*Revision in the manuscript:*

*Page 8, Line 10: "The TVOCs were defined as the sum of the 22 VOC species listed in Text S1. Note that not all detected VOCs were included in this study because of high rates of missing data."*

*Text S1 & Table S1 & Table S2 & Table S4 & Table S10: p-xylene has been added.*

**Comment 1.3**

*Given the extensive annual VOC information available at other sites (local and regional) it would be useful to compare seasonally averaged measurements at as many different sites as possible (in a Table?).*

**Response:**

As suggested, a comparison of seasonally averaged total VOCs (TVOC) at five different sites (1 background, 2 suburban, 1 urban and 1 roadside sites) in Hong Kong has been conducted. The results are presented in Table S8. The table demonstrates the seasonal trends of TVOC were similar at these different sites, with highest values in winter and lowest in summer, though the TVOC levels varied over the sites. In addition, as expected, the TVOC levels were higher at urban and roadside sites, while lower at suburban sites.

*Revision in the manuscript:*

*Supplementary:*

Table S8. Seasonal averages of TVOC ($C_3$-$C_8$) in 2013 at five sites in Hong Kong.

|  | Hok Tsui (Background site) | HKUST (Suburban site) | Tung Chung (Suburban site) | Yuen Long (Urban site) | Mong Kok (Roadside site) |
|---|---|---|---|---|---|
| Spring | 5.9±0.2 | 2.6±0.2 | 4.1±0.2 | 6.3±0.4 | 11±0.5 |
| Summer | 4.5±0.2 | 1.5±0.2 | 2.5±0.2 | 2.9±0.2 | 11±0.3 |
| Autumn | 7.6±0.2 | 2.7±0.1 | 4.7±0.2 | 7.3±0.3 | 16.1±0.4 |
| Winter | 13.6±0.4 | 6.7±0.3 | 7±0.3 | 16±0.7 | 24.6±0.7 |

Note: Hok Tsui is a well-known regional background site at the southeastern tip of Hong Kong. The Hong Kong University of Science and Technology (HKUST) is an Air Quality Research Supersite located on the shorefront of the HKUST campus in the Hong Kong suburban area. Yuen Long (YL) is a typical urban site adjacent to main traffic roads and surrounded by residential and industrial blocks. Mong Kok is a typical roadside site with high traffic density.

*Page 10, Line 6: "Generally, all precursors showed low values in summer and high levels in winter, mainly due to typical Asian monsoon circulations, which brought in clean marine air in summer and delivered pollutant-laden air from mainland*

*China in winter (Wang et al., 2009). Subject to this reason, a similar seasonal variation was observed for the averages of TVOC at different locations over Hong Kong (see Table S8)."*

**Comment 1.4**

*Difficult to separate the background and regional effects on observed ozone formation from the local chemical production and more detail into how the authors have done this is needed.*

The authors agree that it is indeed difficult to separate regional effects on $O_3$ observation from local production due to two assumptions: (1) the background/regional effects we deducted at HT or TM may not be the same as those at TC (TC may have higher or lower regional air than at HT/TM), or (2) HT/TM is also affected by local air mass (what we deducted as background/regional air may contain local emissions). However, we think the approach we used in this study is acceptable and did give reasonable results. To make it clearer, more details on the method have been provided in the revised manuscript. The sensitivity and uncertainty associated with this approach have also been investigated and the results are discussed in Responses to Comment 1.5 and 1.14.

*Revision in the manuscript:*

*Page 6, Line 7:*

*" The measured precursors (i.e., VOCs, NO and $NO_2$) at TC are a mixture of regional background values augmented by local source influences, and the two parts are very difficult to be fully separated. It is worth noting that the regional background values are those observed at locations where there is little influence from urban sources of pollution, while the baseline values mentioned in Section 3.2 are observations made at a site when it is not influenced by recent, locally emitted or produced pollution (TF HTAP, 2010). To minimize the influence of regional transport from the inland PRD region, the real-time regional background values in this study were simply subtracted off from the observations at TC. …… In this study, mixing ratios of 21 anthropogenic VOC species with relatively long lifetimes (5h – 14d) at HT were selected as the background values for deducting from the observed data at TC. The lifetimes of these VOCs were estimated based on the reactions with OH radicals (Simpson et al., 2010). Isoprene was considered as not having a regional impact due to its short lifetime (1-2 h) (Ling et al., 2011). Furthermore, the lifetime of $NO_2$ is determined by the main sinks of $OH+NO_2$ reaction*

*and the hydrolysis of $N_2O_5$ at the surface of wet aerosols, which highly depends on meteorological conditions, such as temperature and humidity (Dils et al. 2008; Evans and Jacob, 2005). Previous experimental studies showed an exponential relationship between the $NO_2$ lifetime and temperature (Dils et al., 2008; Merlaud et al., 2011; Rivera et al., 2013), which was used to estimate the lifetime of $NO_2$ in this study."*

**Comment 1.5**

*What are the sensitivities in the model to these (potentially high) uncertainties in initial concentrations of ozone precursor?*

**Response:**

Thanks for raising this issue. The uncertainties in initial concentrations of ozone precursors are considered into two parts: uncertainties caused by measurement data themselves, and uncertainties caused by the deduction of background/regional values. For the first part, the upper and lower limits of initial precursor concentrations with uncertainties ($\pm$95% confidence intervals, see Table S4) have been used for testing the sensitivity in the model. For this test, a small dataset with randomly selected days in 2012 is used. The sensitivity results are shown in Figure S4 and Figure S5 for model runs with varying $NO_2$ only or VOCs only, respectively. As it is believed that $NO_2$ had little influence on Hong Kong in summer and autumn through regional transport (see Page 6 in the original manuscript for the reasoning), the sensitivity tests of $NO_2$ were only conducted for spring and winter. For the sensitivity tests of VOCs, only the variations of anthropogenic VOCs were considered as the biogenic VOCs have much shorter lifetimes and are considered as negligible source of regional transport in this study. The results demonstrate that the variations in the modelled $O_3$ production are about $\pm$13% and $\pm$3.9% for $NO_2$ and VOCs, respectively, when the precursors change in the range of $\pm$95% confidence intervals.

In addition, the changes in ozone precursors, which are caused by removing their regional background concentrations and are considered as the maximum uncertainties in initial concentrations, have also been investigated by modelling tests. Three scenarios (i.e. removing $NO_2$ only, removing VOCs only, and removing both $NO_2$ and VOCs) are included. Please refer to Response to Comment 1.14 for details.

***Revision in the manuscript:***

*Page 8, Line 1: "Furthermore, the sensitivities in the OBM model to the uncertainties in initial concentrations of ozone precursors have been examined by running the model with varying $NO_2$ or VOCs initial concentrations in the range of ±95% confidence intervals, respectively. The results demonstrate that the modelled $O_3$ production was more sensitive to $NO_2$ than VOCs, with a percentage variation about ±13% and ±3.9%, respectively (see Table S4, Figure S4 & Figure S5)."*

**Table S4**. Means (±95% confidence intervals) of the initial concentrations of ozone precursors used in the OBM model (unit: ppbv for $NO_2$ and pptv for VOCs)

| Precursors | Spring | Summer | Autumn | Winter |
|---|---|---|---|---|
| **$NO_2$** | 6.73 ±1.8 | 5.89 ±1.71 | 5.56 ±1.66 | 8.18 ±2.24 |
| **Propane** | 666.33 ±22.88 | 193.75 ±18.63 | 841.16 ±22.11 | 1444.59 ±30.14 |
| **Propene** | 68.65 ±3.24 | 18.77 ±2.09 | 62 ±3.82 | 175.59 ±6.13 |
| ***n*-Butane** | 331.36 ±18.23 | 175.87 ±17.13 | 399.96 ±15.83 | 682.38 ±25.07 |
| ***i*-Butane** | 253.15 ±11.51 | 116.7 ±11.27 | 309.77 ±10.98 | 549.66 ±17.08 |
| **1-Butene** | 6.73 ±0.78 | 3.03 ±0.76 | 20.96 ±1.98 | 26.59 ±2.49 |
| ***trans*-2-Butene** | 6 ±0.88 | 3.04 ±0.77 | 5.76 ±1.06 | 15.71 ±1.75 |
| ***cis-2*-Butene** | 4.14 ±0.57 | 2.7 ±0.72 | 24.23 ±2.09 | 17.9 ±1.8 |
| **1,3-Butadiene** | 6.49 ±0.83 | 2.22 ±0.54 | 11.97 ±1.76 | 22.05 ±1.52 |
| ***n*-Pentane** | 115.98 ±7.74 | 64.34 ±7.72 | 121.87 ±5.9 | 177.07 ±6.48 |
| ***i*-Pentane** | 192.06 ±9.5 | 122.53 ±12.05 | 228.66 ±8.02 | 300.24 ±8.96 |
| ***n*-Hexane** | 138.87 ±8.98 | 53.85 ±5.63 | 87.7 ±6.28 | 295.35 ±19.74 |
| ***i*-Hexane** | 36.22 ±5.66 | 18.72 ±5.39 | 50.39 ±9.4 | 84.59 ±7.11 |
| ***n*-Heptane** | 24.38 ±2.47 | 12.73 ±1.44 | 16.45 ±3.31 | 41.84 ±5.28 |
| **Benzene** | 279.94 ±8.35 | 68.04 ±2.88 | 399.86 ±7.69 | 872.04 ±20.71 |
| **Toluene** | 497.33 ±22.41 | 114.85 ±11.1 | 657.38 ±23.93 | 1337.9 ±52.09 |
| **Ethylbenzene** | 80.38 ±5.82 | 18.98 ±2.56 | 119.87 ±6.76 | 246.93 ±12.52 |
| ***m/p*-Xylene** | 78.13 ±7.85 | 20.57 ±3.1 | 161.13 ±11.08 | 316.45 ±19.02 |

| | | | | |
|---|---|---|---|---|
| *o*-**Xylene** | 23.14 ±2.81 | 9.46 ±1.85 | 40.92 ±3.31 | 85.85 ±5.89 |
| **1,2,3-Trimethylbenzene** | 1.72 ±0.2 | 1.96 ±0.3 | 1.51 ±0.01 | 2.55 ±0.42 |
| **1,2,4-Trimethylbenzene** | 2.85 ±0.43 | 3.22 ±1.12 | 3.17 ±0.57 | 8.28 ±1.31 |
| **1,3,5-Trimethylbenzene** | 1.91 ±0.2 | 1.81 ±0.22 | 1.71 ±0.18 | 1.88 ±0.26 |

[Figure]

**Figure S4.** Comparison of locally-produced $O_3$ (simulated) at TC site in spring and winter with input of varying initial concentrations of $NO_2$ by subtracting the mean background $NO_2$ (red line), the mean background $NO_2$ plus 95% confidence intervals (C.I, blue line), and the mean background $NO_2$ minus 95% C.I (green line).

[Figure]

**Figure S5.** Comparison of locally-produced $O_3$ (simulated) at TC site over the year with input of varying initial concentrations of VOCs by subtracting the mean background VOCs (red line), the mean background VOCs plus 95% confidence intervals (C.I, blue line), and the mean background VOCs minus 95% C.I (green line).

5 **Comment 1.6**

*Set the overall air quality picture in HK in perspective - How do the observed precursor trends and ratios (e.g. $NO_2/NO_x$) compare to other high population industrial urban areas of the world on a year by year seasonal basis? There is a wide range of comparable data in the literature.*

**Response:**

10 Thanks for the nice suggestion. Regional and global comparisons between this study and the literature have been added in a few places to set the overall air quality picture in HK.

*Revision in the manuscript:*

*Page 8, Line 18: "...indicating increasing $O_3$ pollution in Hong Kong. This finding is consistent with other big cities and regions in the world, such as Beijing (Tang et al., 2009), west plains of Taiwan (Chou, et al., 2006), and Osaka (Itano et al., 2007)."*

*Page 8, Line 19: "The annual average $O_3$ concentration in Hong Kong has increased by 0.56 ppbv $yr^{-1}$ in 2004-2015 which is close to that reported for Osaka (0.6 ppbv $yr^{-1}$) in 1985-2002, and in agreement with Lin et al. (2017) who found the annual mean ozone over Hong Kong increased by about 0.5 ppbv $yr^{-1}$ over 2000-2014. "*

*Page 8, Line 23: "The decreasing trends of NOx and CO, also observed in many other high population industrial urban areas (Geddes et al., 2009; Tang et al., 2009), suggest effective reduction of local emissions from transportation, power plants and other industrial activities (HKEPD, 2016)"*

*Page8, Line 26: "Unlike $O_3$ and $NO_x$, the trend of TVOCs varied across different areas, for example, increasing in Beijing (Tang et al., 2009), decreasing in Toronto (Geddes et al., 2009) and Taiwan (Chou et al., 2006), while almost remained unchanged (p > 0.05) for Hong Kong (Figure 2)."*

***Reference:***

*Lin, M., W. Horowitz, R. Payton, A.M. Fiore, G. Tonnesen (2017). US surface ozone trends and extremes from 1980 to 2014: Quantifying the roles of rising Asian emissions, domestic controls, wildfires, and climate. Atmos. Chem. Phys., doi:10.5194/acp-17-2943-2017.*

**Comment 1.7**

*Difficult to relate specific plots with specific trends discussed in the main text.*

**Response:**

Thanks for pointing out this inconvenience. Efforts have been made to improve the description of each figure caption and cite the plots as many as needed.

**Comment 1.8**

*Confusing in parts and difficult to read/follow.*

**Response:**

Not sure which parts the referee specified. However, the readability of the revised manuscript has been greatly improved with help from English native speakers.

**Comment 1.9**

*Missing reactivity – This study is linked to the measurement of 21 VOC precursors, which may be the main source of ozone in the region in the inventories (show this?). However, how much primary reactivity is potentially missing (e.g. long chain/branched alkanes and alkenes?), on a year by year basis? Some discussion on the missing reactivity and how this may affect the presented results is warranted.*

**Response:**

Thanks for pointing out this important issue. Although much more number of VOC precursors could be quantified in Hong Kong (*i.e.*, over 60+ NMHC species reported in our previous studies), only 21 VOC species (actually 22, please see the Response to Comment 1.2 for reasons) were available for the trend study subject to the capability of the online VOC analyser used at TC site and the completeness of the dataset over the years of interest.

The authors admit that the limited number of VOC precursors would cause missing of reactivity. Luckily, individual VOC species have different photochemical reactivity, and the selected VOCs in this study covered the majority of the total reactivity. Guo et al. (2004) calculated the individual reactivity of VOC species (including 97 hydrocarbons) at two sites in Hong Kong and found that the total reactivity of 21 hydrocarbons (*i.e.*, 1-butene, *i*-butene, isoprene, propene, toluene, *m*-xylene, *p*-xylene, *trans*-2-butene, *n*-butane, *cis*-2-butene, *i*-pentane, 2-methylpentane, hexane, *i*-butane, propane, *n*-pentane, heptane, ethylbenzene, 3-methylpentane, *o*-xylene, and benzene) contributed over 80% reactivity by OH radicals. Compared to the present study, only 1-butene, *i*-butene and 3-methylpentane from this list were not included, leading to about 27% reactivity missing. In addition, it was found that carbonyls would also contribute to the total photochemical reactivity (Cheng

et al., 2010). The lack of carbonyl species may also cause some reactivity missing. This information has been added into the text to state the limitation of this study.

**Reference:**

Cheng, H., et al. (2010). "On the relationship between ozone and its precursors in the Pearl River Delta: application of an observation-based model (OBM)." Environmental Science and Pollution Research 17(3): 547-560.

Guo, H., et al. (2004). "Characterization of hydrocarbons, halocarbons and carbonyls in the atmosphere of Hong Kong." Chemosphere 57(10): 1363-1372.

**Revision in the manuscript:**

*Page 8, Line 11: "The limited number of VOC precursors would cause missing of reactivity which was estimated < 30% for total hydrocarbons based on our previous study (Guo et al., 2004). The missing reactivity would increase if carbonyls are considered (Cheng et al., 2010)."*

*Technical comments*

**Comment 1.10**

*Fig 1 – "cities" of regions? A more detailed map/maps of the local and regional environs including distance to downtown HK would be useful. What are the prevailing wind directions in each season?*

**Response:**

Sorry for the confusing. Although many cities were labelled on the map (Figure 1), we did not mean that "cities" are "regions". As mentioned in the text, the PRD region consists of nine cities, two of which (*i.e.*, Guangzhou and Shenzhen) have a large population over 10 million. To minimise the confusing, only the two megacities are labelled on the revised map. In addition, a sub-map has been supplied to show more detailed environmental information around Hong Kong.

The prevailing wind directions at the measurement site in each season are summarised in Figure S1. As seen in the graph, east wind is the most prevailing in spring and autumn with southwest winds for summer and northeast for winter. This information has been briefly introduced in the revised manuscript with the reference being cited.

*Revision in the manuscript:*

*Section 2.1: Site description*

*Page 3, line 14-15: "The sampling site (22.29 °N, 113.94 °E) is located at about 24 km southwest of downtown Hong Kong and about 3 km south of the Hong Kong International Airport (Figure 1)"*

[Figure]

**Figure 1.** Location of the sampling sites and surrounding environments. Guangzhou and Shenzhen are the two biggest cities in the inland PRD region with a population over 10 million for each city. Hok Tsui (HT) and Tap Mun (TM) are regional background sites. The Hong Kong University of Science and Technology (HKUST) is an Air Quality Research Supersite located in suburban area. Yuen Long (YL) is a typical urban site adjacent to main traffic roads and surrounded by residential and industrial blocks. Mong Kok is a typical roadside site with high traffic density.

*Supplementary:*

[Figure]

**Figure S1**. Wind rose plots showing prevailing winds in Hong Kong for each season. The daily data of 2005-2014 is obtained from the Hong Kong airport (https://www.wunderground.com). Colour indicates wind speed (unit: km h$^{-1}$).

*Page 3, Line 17: "At TC, the prevailing wind varies by seasons, with east winds for spring and autumn, southwest winds for summer and northeast for winter (see Figure S1)."*

**Comment 1.11**

*Page 5 (modelling) - Reference for TUV photolysis modelling needed – model sensitivity to cloud and haze days? Loss of*
10 *radicals to aerosol sensitivity studies? MCM is not an "explicit" mechanism. It is described as "semi-explicit" – more complex and complete chemistry than CB05.*

**Response:**

Selected references have been cited in the text to provide the readers more information on the sensitivity of TUV photolysis model to cloud and haze days, as well as radical loss to aerosol studies. With regard to MCM, the description about its mechanism `has been corrected from "explicit" to "near-explicit" as the reviewer suggested.

*Revision in the manuscript:*

*Page 5, Line 13: "... the photolysis rates of different species in the OBM model was determined using the output of the Tropospheric Ultraviolet and Visible Radiation model (TUV v5) (Madronich and Flocke, 1999) based on the actual conditions of Hong Kong, i.e., meteorological parameters, location, and time period of the field campaign."*

*Page 5, Line 16:" However, it is noteworthy that the atmospheric physical processes (i.e., vertical and horizontal transport), the deposition of species, and the radical loss to aerosol (George et al., 2013; Lakey et al., 2015) were not considered in the OBM model."*

*Page 5, Line 30: "Since the MCM utilizes the near-explicit mechanism describing the degradation of 143 primary VOCs and contains around 16,500 reactions involving 5,900 chemical species…"*

*Reference:*

*George, I. J., et al. (2013). "Measurements of uptake coefficients for heterogeneous loss of $HO_2$ onto submicron inorganic salt aerosols." Physical Chemistry Chemical Physics 15(31): 12829-12845.*

*Lakey, P. S. J., et al. (2015). "Measurements of the $HO_2$ Uptake Coefficients onto Single Component Organic Aerosols." Environ Sci Technol 49(8): 4878-4885.*

*Madronich S, Flocke S. 1999. The Role of Solar Radiation in Atmospheric Chemistry [M] // Boule P. Handbook of Environmental Chemistry. New York: Springer-Verlag, 1–26.*

**Comment 1.12**

*Why was a simple BLH/deposition sensitivity run not performed?- sensitivity?*

**Response:**

Thanks for the nice suggestion. Unfortunately, the module of BLH/deposition is not included in the OBM model used in this study, as mentioned in the original manuscript (Page 5, Line 16). This factor will be incorporated into the model in the future.

**Comment 1.13**

5 *Did the model runs include a "spin up" time to get the radical intermediates into steady state? i.e. run for 2-4 consecutive days and then taking the data for the last 24 hours? This could significantly affect the model $O_3$ coming from the model secondary chemistry and the RIR calculations if the intermediates are not in steady state.*

**Response:**

We admit that the addition of a "spin up" time would get the radical intermediates steady quicker and reduce the
10 modelling uncertainty. However, the influence might be limited for modelling the $O_3$ production. According to a study by Ren *et al.* (2013), running the model for 24 hours would be enough to allow most calculated reactive intermediates to reach steady state.

To further understand such an influence in this study, sensitivity tests have been conducted with a small dataset, which includes randomly selected four days from each season in 2012, with or without a 4-day "spin up" time. As seen in Figure
15 S2, the results demonstrate that the method without the "spin-up" time (red line) indeed causes slight underestimation of the $O_3$ production with a maximum of 4.63% in spring and a minimum of 1.52% in winter.

The influence of "spin-up" time on RIR calculations has also been tested using the same small dataset. It shows that our approach (without spin-up time) systematically underestimates the RIR values, but it does not change the results in each season (Figure S3). In addition, it seems that the RIR of TVOCs is more sensitive to this method. This statement has been
20 added into the text.

[Figure]

**Figure S2.** Comparison of $O_3$ production simulated by with (blue line) or without (red line) a 4-day "spin-up" time in different seasons. In the tests, four days data were randomly selected from each season in 2012 as an example. The results demonstrate that the method without the "spin-up" time indeed causes slight underestimation of the $O_3$ production with a maximum of 4.63% in spring and a minimum of 1.52% in winter.

[Figure]

**Figure S3.** Comparison of RIR calculated by with (solid bar) or without (light bar) a 4-day "spin-up" time in different seasons. In the tests, one-day data was randomly selected from each season in 2012 as an example. It shows that the approach (without spin-up time) systematically underestimates the RIR values, but it does not change the results in each season. In addition, it seems that the RIR of TVOCs is more sensitive to this method.

*Revision in the manuscript:*

*Page 5, Line 18: "In addition, a "spin-up" time was not applied in the model to get the radical intermediates well steady which might have caused a slight underestimation on the simulated $O_3$ production (Figure S2) and its sensitivity to precursors (Figure S3)."*

*Reference:*

*Ren et al., 2013. Atmospheric oxidation chemistry and ozone production: Results from SHARP 2009 in Houston, Texas. Journal of Geophysical Research: Atmospheres 118(11), 5770-5780.*

**Comment 1.14**

*The reasoning behind the removal of "background" and "local" concentrations to give an "observed" value are vague and questionable. (1) Their influence will be highly variable with respect to the meteorology and time of the year. (2) What are these concentrations and how were they derived (by wind direction? hourly, monthly, seasonally?). (3) Again, I would like to see the **seasonal concentrations** of VOC measurements at other regional and local sites compared and contrasted to the current data set in some detail and (4) how sensitivity to this calculation effects the deconvolution of local photochemical ozone formation. (5) The authors state that the "NO₂ emitted from the inland PRD is slightly more likely (?) to arrive at TC in winter and spring than in summer and fall, the differences in travel time among the seasons are relatively small and it is difficult to be precise with seasonal average estimates of NO2 lifetime and travel time". Therefore, the errors and limitations in understanding the effect of regional/background NO2 on the observed O3 are high here (which the authors acknowledge) – what are the uncertainties associated with this and what is the sensitivity to the model runs?*

**Response:**

Thanks for the comments. However, we get a bit confused as we did not remove the "background" and "local" concentrations to get an "observed" value. Actually, what we did was to get "pure" local concentrations of precursors by subtracting the "background" concentrations, which were measured at a background site and were assumed being highly influenced by regional transport, from the observations at the receptor site. Then the calculated "local" concentrations were used for simulation of locally-produced $O_3$. This confusing on the method description has been resolved by changing the wording (see Response to Comment 1.4).

(1) Although the meteorology and time of the year are variable, the influence of background removal approach on the results would be minor for the hourly observation at the regional sites was considered as "regional background" and deducted from the data observed at the receptor site (TC). In other words, the variable meteorological and seasonal factors had already been considered in our method.

(2) As mentioned above, the concentrations obtained by subtracting off the regional background from the observations are the "pure" locally-produced concentrations. They were derived by hourly observations at the regional site and at the receptor site.

(3) The seasonal concentrations of VOC measurements at five different sites (namely HT, TC, YL, MK and HKUST) are summarised in Table S8. It shows that the seasonal variations are quite similar though their levels are different. Please see Response to Comment 1.3 for details.

(4) The authors are a bit confused by this comment, as the deconvolution of local photochemical ozone formation was not discussed in the paper. As mentioned above, the background removal method was applied to get a "pure" locally-produced $O_3$ which was considered as a part of observed $O_3$. For the sensitivity of locally-produced $O_3$ to initial VOC concentrations, please see the Response to Comment 1.5.

(5) As stated in the text, the authors admit that this approach dealing with $NO_2$ may cause uncertainty on simulation of locally-produced $O_3$. The sensitivity tests on initial $NO_2$ levels (see Response to Comment 1.5) also confirm this point. The maximum uncertainties associated with this, for 4 different sites (*i.e.*, HKUST, TC, YL and MK) in Hong Kong, have been assessed by comparing the changes in locally-produced $O_3$ (simulated) before and after removing the background $NO_2$ only (scenario 1) or the background TVOCs only (scenario 2) or both background $NO_2$ and TVOCs (scenario 3). The results for scenarios 1, 2 and 3 are given in Table S5, S6 and S7, respectively. Compared to the removal of VOCs, the removal of $NO_2$ had a much bigger influence on the simulation of locally-produced $O_3$. The influence was higher in spring and winter than in summer and autumn, consistent with our assumption based on calculation of precursor's lifetime: stronger regional transport in spring and winter. Therefore, the results support our approach to remove background $NO_2$ only in spring and winter seasons. In addition, the seasonal variations of the influence are similar across different sites.

Furthermore, we have compared annual trends of local $O_3$ production simulated with original precursor observations and with background removed precursor concentrations (see Figure S6). A similar trend is observed between with and without removing background values. Therefore, our method does not change the conclusion at least, though it might be associated with some degree of uncertainty.

*Revision in the manuscript:*

*Page 8, Line 4: "In addition, the uncertainties associated with removing the background concentrations are also evaluated,*

*suggesting a similar trend for simulated locally $O_3$ production for both approaches (See Figure S6 & Tables S5-S7)."*

*Supplementary:*

5    **Table S5**. The difference of locally-produced $O_3$ (simulated) before and after removing regional background $NO_2$

| | HKUST (Suburban site) | Tung Chung (Suburban site) | Yuen Long (Urban site) | Mong Kok (Roadside site) |
|---|---|---|---|---|
| Spring | | -33.1% | -33.1% | -11.7% |
| Summer | -8.3% | -12.9% | -7.5% | -2.6% |
| Autumn | -14.8% | -17.7% | -13.9% | -5.6% |
| Winter | -26.6% | -46.0% | -47.9% | -21.9% |

Note: Online VOC data was available at five sites from April 2011 to January 2012; Only four months trace gases data at HKUST site were available in 2011; Totally, the simulation days at TC, YL, MK and UST are 286, 273, 233 and 105 days.

**Table S6.** The difference of locally-produced $O_3$ (simulated) before and after removing regional background VOCs

| | HKUST (Suburban site) | Tung Chung (Suburban site) | Yuen Long (Urban site) | Mong Kok (Roadside site) |
|---|---|---|---|---|
| Spring | | -2.6% | -2.6% | -1.2% |
| Summer | -2.4% | -2.7% | -2.1% | -0.3% |
| Autumn | -1.8% | -1.8% | -4.0% | -3.0% |
| Winter | -0.6% | -4.0% | -4.8% | -4.5% |

Note: Online VOC data was available at five sites from April 2011 to January 2012; Only four months trace gases data at HKUST site were available in 2011; Totally, the simulation days at TC, YL, MK and UST are 286, 273, 233 and 105 days.

**Table S7**. The difference of locally-produced $O_3$ (simulated) before and after removing both regional background $NO_2$ and VOCs.

|  | HKUST (Suburban site) | Tung Chung (Suburban site) | Yuen Long (Urban site) | Mong Kok (Roadside site) |
|---|---|---|---|---|
| Spring |  | -33.5% | -33.2% | -12.6% |
| Summer | -9.8% | -13.7% | -7.9% | -2.7% |
| Autumn | -15.5% | -17.8% | -14% | -6.0% |
| Winter | -27.1% | -46.2% | -48.4% | -22.1% |

Note: Online VOC data was available at five sites from April 2011 to January 2012; Only four months trace gases data at HKUST site were available in 2011; Totally, the simulation days at TC, YL, MK and UST are 286, 273, 233 and 105 days.

[Figure]

**Figure S6.** Annual trends of simulated local $O_3$ production (blue line: with background precursors; red line: without background precursors) in four seasons at TC during 2005–2014. Error bars represent 95% confidence interval of the averages.

**Comment 1.15**

*Eq 7 – delta kro2+no[RO2][NO]?*

**Response:**

Thanks for pointing out this error.

10 ***Revision in the manuscript:***

*Page 7, Line 17:* The equation 7 has been revised as follows:

$$G^S_{O_3\text{-}NO} = k_{HO_2+NO}[HO_2][NO] + \sum k_{RO_{2i}+NO}[RO_{2i}][NO]$$

**Comment 1.16**

*Fig 3. Not sure how significantly different the inter-annual trends of the TVOC in spring and autumn are?*

**Response:**

As shown in Figure 3, the inter-annual trends of TVOCs in spring and autumn significantly decreased, however it is difficult to quantify how significantly different these trends were in the two seasons. What we can only conclude is that the TVOCs in autumn decreased slightly quicker than those in spring as the decreasing rates were -0.32 and -0.26, respectively. In addition, the inter-annual trends of the TVOC reactivity in spring showed no significant change, while the trends in autumn decreased significantly (see Figure S16), indicating some sort of difference in VOC concentration and composition in the two seasons. Please refer to line 23, page 12 for details.

**Comment 1.17**

*Fig 4. Why are the locally produced simulated and filtered trends so different in most seasons?*

Response:

The two types of locally-produced $O_3$ were obtained by two different approaches, that is, photochemical simulation with "pure" local precursors and simple filtering by wind speed. So it is not surprising to see a difference between the trends of the simulated and filtered data, though theoretically higher percentage of locally-produced $O_3$ observed at lower wind speed. On the other hand, both results were at a similar level, indicating that they were comparable.

**Comment 1.18**

*Page 10 – "The different inter-annual trends of $NO_x$ and TVOCs in spring/autumn from those in summer/winter were probably because marine air significantly diluted air pollution in summer while continental air masses remarkably burdened*

*air pollution in winter, which concealed the decreased local emissions of $NO_x$ and TVOCs in summer and winter (Wang et al., 2009)" – do you have winder sector data to show evidence for this?*

**Response:**

Wind rose plots for each season have been given in Figure S1 which shows more southwest winds (*i.e.*, marine air) in summer and more northeast winds (*i.e.*, continental air) in winter. Please refer to the Response to Comment 1.10 for more information.

**Comment 1.19**

*Define and show NO titration reaction as a separate Equation.*

**Response:**

This comment has been well accepted.

*Revision in the manuscript:*

*Page 10, Line 27:* The following reactions related to NO titration have been added in the text.

$$NO_2 + hv \rightarrow NO + O \tag{R1}$$
$$O + O_2 + M \rightarrow O_3 + M \tag{R2}$$
$$NO + O_3 \rightarrow NO_2 + O_2 \tag{R3}$$

*Page 10, Line 21: "Here the NO titration refers to the ''titration reaction'' between NO and $O_3$. Although $NO–NO_2–O_3$ reaction cycling (including the effects of NO titration, see reactions R1-R3) can be theoretically regarded as a null cycle and provides rapid cycling between NO and $NO_2$, the NO titration effect can retard the accumulation of $O_3$ in an urban environment by means of substantial NO emissions (Chou et al., 2006)."*

**Comment 1.20**

*Further investigation found that temperature and solar radiation in summer indeed increased in these years (p<0.05)" Show this data.*

**Response:**

This data was included in the original supplementary. Please refer to Figure S9 for details.

*Revision in the manuscript:*

*Page 10, Line 27: "Further investigation found that temperature and solar radiation in summer indeed increased (p<0.05)*

*in these years (see Figure S9), whereas they had no significant change in other seasons (the reasons remained unclear),…"*

**Comment 1.21**

5  *Page 11 – Section 3.2 repeats itself somewhat*

**Response:**

This comment has been well accepted, and the first sentence has been revised to minimise the confusing.

*Revision in the manuscript:*

*Page 12, Line 2: "In this study, the OBM (CB05) model was used to simulate the long-term trends of $O_3$ produced by in-situ*

10  *photochemical reactions (hereinafter locally-produced $O_3$ (simulated))."*

**Comment 1.22**

*Page 12 – the MCM is not "explicit". Should also be (p<0.05)?*

**Response:**

Thanks for the suggestion. In the text (page 5, line 31), "explicit" has been replaced by "near-explicit" to describe the

15  mechanism applied in MCM.

For the *p*-value, though the MCM considers much more detailed chemical reactions than CB05, the simulated temporal

patterns of $O_3$ were similar, which should be. So we do not think that "*p>0.05*" in the sentence, "*because the simulated*

*results of both CB05 and MCM models followed similar temporal patterns (p>0.05)*", should be changed to "*p<0.05*".

**Comment 1.23**

20  *"locally-produced $O_3$ (filtered) values clearly showed similar trends to locally-produced $O_3$ (simulated) in spring, autumn*

*and winter (p=0.07, 0.09 and 0.93, respectively)" - where is the plot showing these trends?*

**Response:**

Sorry for this confusing. Please refer to Figure 4 for these trends. In addition, we have realised that these *p*-values were used inappropriately. They only suggest the average levels of the type of data in these seasons were similar. This error has been fixed in the revised manuscript.

*Revision in the manuscript:*

5    *Page 12, Line 27: "locally-produced $O_3$ (filtered) values clearly showed similar trends to locally-produced $O_3$ (simulated) in spring, autumn and winter (see Figure 4)"*

**Comment 1.24**

*"Unlike in spring, though the observed and locally-produced O3 (filtered) displayed increasing trends in summer (0.70±0.34 ppbv/yr and 0.66±0.41 ppbv/yr, respectively; p<0.05)," – these trends are not the same as shown in Fig 4?*

10  **Response:**

Sorry for this error. Only the values in Figure 4 are correct. This inconsistence problem has been resolved.

*Revision in the manuscript:*

*Page 13, Line 4: "Unlike in spring, though the observed and locally-produced $O_3$ (filtered) displayed increasing trends in summer (0.67±0.34 ppbv/yr and 0.61±0.41 ppbv/yr, respectively; p<0.05)..."*

15  **Comment 1.25**

*Page 13 – "the total solar radiation (0.24±0.16 MJ·m-2yr-1, p<0.01) and temperature (0.095±0.034 oC/yr, p<0.05) in summer significantly increased during the past 10 years" – odd units! Solar radiation should be given as irradiance in W m-2. Temperature in K. Be consistent as to how you write units – K/y or K y-1. Why has the solar radiation increased in the summer over 10 years? Less haze?*

20  Response:

Thanks for pointing out these issues. The unit of temperature has been changed to "K". However, MJ·m$^{-2}$ is a unit for the integration of solar radiation in a certain amount of time. This unit has been widely used in literature to show a general level

of solar radiation. More important, the only solar radiation data available in this study are recorded in this unit and we have no way to convert them back to data with a unit of "W m$^{-2}$".

The statement on the increase of solar radiation in summer along the 10 years was based on measurements from Hong Kong Observation, but frankly speaking, the reasons remain unknown. It might be due to less haze as the air quality has been getting better in Hong Kong and the PRD. To make this point clear, the text has been improved accordingly.

*Revision in the manuscript:*

*Page 13, Line 8: "Indeed, the total solar radiation (0.24±0.16 MJ m$^{-2}$ yr$^{-1}$, p<0.01) and temperature (0.095±0.034 K yr$^{-1}$, p<0.05) in summer significantly increased during the past 10 years (see Figure S9), subsequently resulting in the enhanced in-situ photochemical reactivity of VOCs, although their quantitative contributions remain unknown and require further investigation. The increase of solar radiation might be due to the decreasing haze as the air quality has been getting better in Hong Kong and the PRD (Louie et al., 2013)."*

*Reference:*

*Louie, P. K. K., et al. (2013). "A Special Issue of Atmospheric Environment on "Improving Regional Air Quality over the Pearl River Delta and Hong Kong: From Science to Policy" Preface." Atmospheric Environment 76: 1-2.*

**Comment 1.26**

*Fig 9. Why is the data not presented as monthly averages or even by season – which would be clearer and possibly give more information?*

**Response:**

Thanks for this suggestion. Figure 9 has been redrawn to make it clearer and more informative. The values have also been updated accordingly.

[Figure]

Figure 9. Trends of the daytime averaged contribution of four VOC groups to $O_3$ mixing ratio: (a) AVOC (Aromatics), (b) AVOC (Alkenes), (c) AVOC (Alkanes) and (d) BVOC at TC during 2005–2014.

*Revision in the manuscript:*

*Page 21, Line 34: "Unlike AVOC (Aromatics/Alkanes), the contribution of AVOC (Alkanes) to $O_3$ formation during 2005-2014 showed no significant change (p=0.23), ..."*

*Page 22, Line 4: "Furthermore, BVOC showed no evident change in the contribution to $O_3$ mixing ratios during the last*

10 *decade (p=0.57), ..."*

**Comment 1.27**

*Page 20 (contribution of VOC groups) – I would have liked to see the year by year seasonal trends of the individual TVOC groups plotted – i.e. aromatics, alkanes, alkenes and BVOC. This would give a more comprehensive overview of the TVOC trend and how it evolved/is evolving.*

5 **Response:**

Thanks for the suggestion. A new graph has been supplied in the supplementary to show the year by year seasonal trends of the individual VOC groups (Figure S18). Text has been added to describe the new plot accordingly.

*Revision in the manuscript:*

10 *Page 8, Line 29: "Moreover, the long-term trends of individual VOCs, except for BVOC, were different from that of TVOCs (see Figure S18) because many control measures were taken in the last decade, which altered the composition of VOCs in the atmosphere, such as the reduction of toluene by solvent usage control and the increase of alkanes in Liquefied Petroleum Gas (LPG) in 2005-2013 (Ou et al., 2015) and the decrease of LPG-alkanes in 2013-2014 (Lyu et al., 2016)."*

15 *Supplementary:*

[Figure]

Figure S18. Trends of monthly average of individual VOC groups (*i.e.*, aromatics, alkenes, alkanes and BVOCs) at TC during 2005-2014.

**Comment 1.28**

5   *Page 21. Basic detail of the HK government emissions reduction plan need to be outlined (with the reference website placed in the References section).*

**Response:**

Thanks for this suggestion. Actually, there were a few places already describing the HK government emissions reduction plan in the original manuscript. For example,

*"In fact, the Hong Kong Government has implemented a series of VOC-control measures since 2007 (HKEPD, 2016). From April 2007, the Air Pollution Control (VOCs) Regulation was implemented to control VOC emissions from regulated products, including architectural paints/coatings, printing inks and six selected categories of consumer products. In January 2010, the regulation was extended to control other high VOC-containing products, namely vehicle refinishing paints/coatings, vessel and pleasure craft paints/coatings, adhesives and sealants."* (Page 21, Line 14-18), and

*"During 2005-2014, the Hong Kong government launched a series of measures to reduce vehicular emissions, including diesel, LPG and gasoline vehicles (http://www.epd.gov.hk/epd/english/environmentinhk/air/prob_solutions/air_problems.html)."* (Page 21, Line 25-26).

To provide more background information to readers, a reference has been added in the revised manuscript.

*Revision in the manuscript:*

*Page 21, Line 14: "In fact, the Hong Kong Government has implemented a series of VOC-control measures since 2007 (HKEPD, 2016)."*

*Reference:*

*HKEPD (Hong Kong Environmental Protection Department): 2014 Hong Kong Emission Inventory Report, available at: http://www.epd.gov.hk/epd/sites/default/files/epd/2014Summary_of_Updates_eng_2.pdf, (available on 2017-06-15).*

**Comment 1.29**

*"photolysis rates of alkanes" – I am sure alkanes do not photolyse in the atmosphere!*

**Response:**

Sorry for the confusing. What we attempted to say is the reaction rate of alkanes with OH radicals.

*Revision in the manuscript:*

*Page 22, Line 2: "In addition, the seasonal variation of $O_3$ formation, of which the reaction rates of alkanes with OH radicals were high in summer and low in winter, would also blur the trend".*

**Responses to Referee 2's comments**

**General comment**

*The authors report an analysis of time series for O3, NOx, TVOCs and CO for Hong Kong for the years 2005-2014. Based on a seasonal analysis of observed and modelled data using an observation-based box model coupled with CB05 mechanism the authors find different trends of these pollutants for each season. Overall, they state that locally produced O3 increased in spring and decreased in autumn over the years. The authors suggest that different decreasing rates in O3 precursors NOx and TVOC as well as changes in VOC composition and/or VOC reactivity (mainly caused by decrease of aromatic compounds) might have led to these O3 trends. For the autumn season the authors state that regional O3 might have been a dominant factor in the O3 trend. An analysis of incremental reactivity showed decreasing contribution from aromatic compounds, while the contribution from alkenes appeared to increase over the years. This might have been due to changing VOC source contributions (less solvents, more traffic emissions). Overall, this paper shows some valuable material and associated discussion. However, there are some important issues which need to be addressed before this paper can be accepted.*

We highly appreciate the reviewer for the positive comments and constructive suggestions. In general, all these comments have been addressed properly with addition of great details of the methods and in-depth discussion in the revised manuscript. Also, many sentences have been reorganised to clear confusing and improve readability. In the following section, the author's responses (in blue) are immediately after the reviewer's comments (in black), with the changes in manuscript at the end (in italic).

**Major issues:**

**Comment 2.1**

*In most figures intra- and inter-annual variations are significantly larger than the 2005-2014 trend. For instance, the O3 trend shows the highest increase from 2005-2014 (0.67 ppb/yr) in autumn (Fig. 3). However, this trend is only determined by 3 "outlier" months of the years 2012, 2013, 2014. These 3 months are just 10% of this specific data set. Another example is*

*Fig. 5 which shows very large scatter in O3 data for the autumn season. Also, for instance Fig. S10 about the annual trends of VOC/NOx ratios is largely determined by the last two years. The question is: how robust are all the trends shown in this paper?*

**Response:**

Thanks for raising this issue. We agree that the validity of the results is very important for a study. For a trend study using statistical techniques, however, it is not surprising to have a larger variation than the trend. In general, the robustness of the trends can be judged by the *p*-value for the regression model. A *p*-value less than 0.05 was considered as acceptable in this study.

The outliers are another issue. What we want to argue here is whether they are real outliers or not. For example, the three extremely high monthly averages of $O_3$ in autumn (see Figure 3), in the point of our view, might be not outliers, because each of them represents one month of data with relative small uncertainty. In other words, each point represents many valid observations during that period. We cannot simply consider them as outliers and exclude them from the dataset. To further clarify this point, monthly averages of $O_3$ observations over 2005-2014, as well as the ratios of TVOCs to NOx, are shown in Figure S10 and Figure S17, respectively.

The Figures have been added into the Supplementary and noted in the main text.

[Figure]

Figure S10. Monthly trend of observed $O_3$ at TC in 2005-2014

[Figure]

Figure S17. Monthly trend of VOCs/NOx ratio at TC in 2005-2014

**Revision in the manuscript:**

*Page 11:* Caption for Figure 3

*"In the sub-plot for O₃ trend in autumn, the three extremely high values are not considered as outliers as each of them represents one month of data with relatively small uncertainty (see Figure S10)."*

*Page 20, Line 2:*

*"Furthermore, the monthly variation of VOC/NO$_x$ ratios showed a significant decreasing trend at a rate of -0.02 yr-1 (p<0.05) (see Figure S17), indicating that VOC reduction became more effective in reducing O₃ in the past 10 years, which is consistent with the conclusions from the above modeling results."*

**Comment 2.2**

*According to HKEPD (2015) long-term trends for O3, NOx, and CO may have been different within the Hong Kong area and not necessarily the same as at the TC site. For O3, annual values at the rural site were highest, but did not change that much over the years, while urban and New Town sites show some increase at overall lower levels than at the rural site. Apart from that NOx values did not change that much for New Town sites, while urban sites indeed showed some slight decrease. For CO there were actually some increases at urban sites over the last years in contrast to New Town sites. The question is: how representative is the TC site for a trend analysis for Hong Kong?*

**Response:**

The authors admit that spatial variations in the levels of O₃ and precursors may be significant across Hong Kong, and that the trends observed at TC might be different with other locations in the city. However, TC site was the best choice for us to select for analysing the trend of O₃ and precursors in Hong Kong. The reasons are as follows:

(1) TC is characterized as a polluted receptor site as it receives urban plumes from Hong Kong and inland PRD region under prevailing north-easterly winds in autumn when most heavy O₃ pollution often occurs in the region. Therefore, the trend of pollutant levels at TC site can generally represent the overall trend of pollutants in Hong Kong with regional impact.

(2) TC site is located in the area of Hong Kong that experiences the most serious O₃ pollution (Zhang et al. 2007; Xue et al., 2014). So understanding the trend of O₃ and precursors at this site will provide critical information to the government for initiation of effective emission reduction plans in future, and

(3) The measurements taken at TC site comprised the most comprehensive source of data, including long-term observations of hydrocarbons, which is very important for a high-quality trend study.

The aforementioned reasons for site selection have been added into the section of site description (section 2.1).

**Comment 2.4**

*Fig. 3 shows some interesting feature. Not only all $O_3$ precursor values are lowest in summer, but also $O_3$ values in summer are lower than in spring and lower than in autumn. They are just slightly higher than in winter. This is a bit astonishing as one would expect highest O3 values in summer. I was wondering whether the authors can shed some light on this and explain the specific summertime conditions.*

**Response:**

Although the solar radiation in summer is generally higher, it is not uncommon to see low $O_3$ levels in summer in a coastal region, like Hong Kong, where the east Asian monsoon brings in clean ocean air masses with less $O_3$ precursors to this area in this season, just like what we described in the original manuscript (Page 10, Lines 6-8):

*"Generally, all precursors showed low values in summer and high levels in winter, mainly due to typical Asian monsoon circulations, which brought in clean marine air in summer and delivered pollutant-laden air from mainland China in winter (Wang et al., 2009)."*

In addition, the high frequency of rainy days in summer was also not in favour of $O_3$ formation. This point has been added in the revised text.

*Revision in the manuscript:*

*Page 10, Line 9: "With lower (diluted) precursor concentrations, together with high frequency of rainy days, it is not uncommon for Hong Kong to see lowest $O_3$ values in summertime (see Figure 3)."*

**Other comments:**

**Comment 2.5**

*Page 4 L5-8: The reference HKEPD (2015) lists various instruments being used in the Hong Kong network. What instruments were actually installed at TC, what were their detection limits, what their resolutions? Was $NO_2$ measured directly?*

**Response:**

The instruments installed at TC site, as well as the detailed description (*i.e.*, detection limit and resolution), have been amended in the method section. In addition, $NO_2$ and $NO_x$ were directly measured by the instrument, while NO was obtained by the difference of $NO_x$ and $NO_2$.

*Revision in the manuscript:*

*Page 4, Line 6:*

*"2.2 Measurement techniques*

*Hourly observations of $O_3$, CO, $SO_2$, $NO$-$NO_2$-$NO_x$ and meteorological parameters at TC from 2005 to 2014 were obtained from the HKEPD (http://epic.epd.gov.hk/ca/uid/airdata). Briefly, $O_3$ was measured using a commercial UV photometric instrument (Advanced Pollution Instrumentation (API), Model 400A) with a detection limit of 0.6 ppbv. CO was measured with a gas filter correlation CO analyser (Thermo ElectronCorp. (TECO), Model 48C) with a detection limit of 0.04 ppm. $SO_2$ was measured using a pulsed fluorescence analyser (TECO, Model 43A) with a detection limit of 1.0 ppbv. $NO$-$NO_2$-$NO_x$ were detected using a commercial chemiluminescence with an internal molybdenum converter (API, Model 200A) and a detection limit of 0.4 ppbv. All the time resolutions for these gas analysers were 1 hour. To ensure a high degree of accuracy and precision, the QA/QC procedures for gaseous pollutants were identical to those in the US air quality monitoring program (http://epic.epd.gov.hk/ca/uid/airdata). The accuracy of the monitoring network was assessed by performance audits, while the precision, a measure of the repeatability, of the measurements was checked in accordance with HKEPD's quality manuals. For the gaseous pollutants, the accuracy and precision within the limits of ±15 and ±20 % were adopted, respectively (HKEPD 2015).*

*Real-time VOC data at TC were also measured by the HKEPD. An online GC-FID analyser (Synspec GC 955, Series 600/800) was used to collect VOC speciation data continuously with a time resolution of 30 minutes. The VOC analyser*

*consists of two separate systems for detection of $C_2$–$C_5$ and $C_6$–$C_{10}$ hydrocarbons, respectively. Detailed description about the real-time VOC analyser can be found in Lyu et al. (2016). Twenty-eight $C_3$-$C_{10}$ VOC species were identified and quantified using this method. In terms of the QA/QC for VOC analysis, built-in computerized programs of quality control systems such as auto-linearization and auto-calibration were used. Weekly calibrations were conducted by using NPL*

5   *standard gas (National Physical Laboratory, Teddington, Middlesex, UK). In general, the detection limits of the target VOCs ranged from 2 to 56 pptv. The accuracy of each species measured by online GC-FID was determined by the percentage difference between measured mixing ratio and actual mixing ratio based on weekly span checks and monthly calibrations. The precision was based on the 95% probability limits for the integrated precision check results. The accuracy of the measurements was about 1-7%, depending on the species, and the measurement precision was about 1-10% (Table*

10   *S1). In addition, the quality of the real-time data was assured by regular comparison with whole-air canister samples collected and analysed by University of California at Irvine (UCI). More details can be found from previous studies in Hong Kong (Xue et al., 2014; Ou et al., 2015; Lyu et al., 2016).*

*For data analysis, linear regression and error bars represented as 95% confidence intervals were used. Trends of $O_3$ and its precursors with a p value < 0.05 were considered significant (Guo et al., 2009)."*

15   **Comment 2.6**

*Page 4 L9: I am surprised to see that only 21 VOCs were identified and quantified at TC given the fact that it is an urban site. Looking into the CO data, which varies between 400 ppb and more than 1 ppm as monthly means (Fig 2), I would expect significantly higher number of VOCs. I doubt the authors can consider the sum of the quantified VOCs as the total VOCs (TVOCs). What do the authors estimate is the fraction of the quantified VOCs on the entire mass of VOCs in ambient*

20   *air at TC?*

**Response:**

The main reason why only 21 (actually 22 in the revised manuscript) species were included in the present study is because of high percentage of missing data for several species at TC site over 2005-2014. As this study looks into the trend

of $O_3$, those species with high missing data were excluded to eliminate the influence caused by different input of VOC species.

The authors admit that the limited number of VOC precursors would cause missing of reactivity. Luckily, individual VOC species have different photochemical reactivity, and the selected VOCs in this study covered the majority of the total reactivity. Guo et al. (2004) calculated the individual reactivity of VOC species (including 97 hydrocarbons) at two sites in Hong Kong and found that the total reactivity of 21 hydrocarbons (*i.e.*, 1-butene, *i*-butene, isoprene, propene, toluene, *m*-xylene, *p*-xylene, *trans*-2-butene, *n*-butane, *cis*-2-butene, *i*-pentane, *2*-methylpentane, hexane, *i*-butane, propane, *n*-pentane, heptane, ethylbenzene, 3-methylpentane, *o*-xylene, and benzene) contributed over 80% reactivity by OH radicals. Compared to the present study, only 1-butene, *i*-butene and 3-methylpentane from this list were not included, leading to about 27% reactivity missing. In addition, it was found that carbonyls would also contribute to the total photochemical reactivity (Cheng et al., 2010). The lack of carbonyl species may also cause some reactivity missing. This information has been added into the text to state the limitation of this study.

In addition, *m*-xylene listed in Tables S1 and S2 actually represents *m/p*-xylene (*i.e.*, *m*-xylene and *p*-xylene) as the VOC analysis method used in this study could not divide the two species apart. To reflect this fact, the number of VOC species adopted in the study has been corrected from 21 to 22. It is noteworthy that this change would not affect our results and conclusion, because the three xylene isomers have the same CB05 model specie (*i.e.*, XYL, see Table S2).

*Revision in the manuscript:*

*Page 8, Line 10: "The TVOCs were defined as the sum of the 22 VOC species listed in Text S1. Note that not all detected VOCs were included in this study because of high rates of missing data."*

*Page 8, Line 11: "The limited number of VOC precursors would cause missing of reactivity which was estimated < 30% for total hydrocarbons based on our previous study (Guo et al., 2004). The missing reactivity would increase if carbonyls are considered (Cheng et al., 2010)."*

*Text S1 & Table S1 & Table S2 & Table S4 & Table S10: p-xylene has been added.*

**Response:**

The accuracy of each species measured by online GC-FID was determined by the percentage difference between measured mixing ratio and actual mixing ratio based on weekly span checks and monthly calibrations. For more details on the method description, please refer to the Response to Comment 2.5 above.

**Comment 2.8**

*Page 5 L25-29: The authors only measured 21 VOCs. What assumptions did the authors have on other VOCs not measured, but needed as an input for MCM?*

**Response:**

It was assumed that the measured VOCs contributed a dominant fraction to $O_3$ production, and that the initial concentrations of those VOCs not measured but needed by MCM were zero. We admit that the use of a limited number of VOCs would cause photochemical reactivity missing. Please refer to Response to Comment 2.6 above for more details on this issue.

**Comment 2.9**

*Page 5 L30-31. It sounds like MCM has been developed by the authors referenced in this sentence. This should be clarified.*

**Response:**

Sorry for this misleading. We admit that the MCM was originally developed by the University of Leeds (http://mcm.leeds.ac.uk/MCM/). Only some developments on localization of the model for Hong Kong and addition of

chemical reaction pathways of more biogenic VOC species and alkyl nitrates have been made by our group. This point has been added in the revised text.

*Revision in the manuscript:*

*Page 6, Line 3: "A more detailed description of the MCM can refer to Jenkin et al. (1997 and 2003) and Saunders et al.*
*(2003). Some developments on localization of the MCM for Hong Kong and addition of chemical reaction pathways of more*
*biogenic VOC species and alkyl nitrates are given in our previous papers (Lam et al., 2013; Cheng et al., 2013; Ling et al.,*
*2014; Lyu et al., 2015)."*

**Comment 2.14**

*Page 6 L21: Do these lifetimes for $NO_2$ include all $NO_2$ relevant reactions or do they refer to just one specific reaction? Please explain why uncertainties show up in these lifetimes. Why were those lifetimes calculated for each season, but not for each day, as the model is run for each day?*

**Response:**

The two types of sink reactions were introduced in the text to provide background information only, and they were not used for the calculation of lifetime values. Actually, the $NO_2$ lifetimes were estimated using the observations and the reported experimental equation as a function of temperature. The uncertainty of the estimated lifetimes resulted from the variation of temperature over time.

In addition, the lifetimes of $NO_2$ were calculated for each day first based on hourly averaged temperatures. Then the seasonal averaged lifetimes were obtained by averaging the daily values.

**Comment 2.15**

*Page 6 L25: Where do the uncertainties in the wind speed calculations come from?*

**Response:**

The uncertainties in the calculation of the transport time from the inland PRD to the TC site came from the 95% confidence

5    intervals of the seasonal average wind speeds.

**Comment 2.16**

*Page 7 L7-8: Please explain how $O_3$ will be produced with titration by NO.*

Response:

Thanks for pointing out this confusing. The sentence has been revised accordingly.

10   *Revision in the manuscript:*

*Page 7, Line 15: "In both runs, $O_3$ production modulated by NO titration is considered during the evaluation period."*

**Comment 2.17**

*Page 7, L16: With regard to the precursors $NO_x$, total VOCs and CO did the authors calculate arithmetic means or medians? Would there be differences?*

15   **Response:**

The levels of $O_3$ and precursors in this study were presented by arithmetic means. We agree that there would be some differences between the results from the two different statistical approaches. However, since the distributions of precursors were near normal distributions due to the relatively large number of samples, the difference in the results from the two average methods was not significantly large. In addition, the application of arithmetic means in this study would make our

20   results easy to compare with other studies.

**Comment 2.18**

*Page 7 L19-20: Please explain whether the monthly maximum $O_3$ level was the monthly averaged daily maximum 8-h $O_3$ average or something else?*

**Response:**

In our manuscript, the monthly maximum $O_3$ level means the maximum of averaged daily maximum 8-h $O_3$ average in one month.

*Revision in the manuscript:*

*Page 8, Line 15: "The monthly maximum $O_3$ level, which was defined the maximum of DMA8 $O_3$ in one month, increased from about 68 ppbv in 2005 to 86 ppbv in 2014, exceeding the ambient air quality standards in Hong Kong (i.e. 80 ppbv)."*

**Comment 2.19**

*Page 8 L6: It sounds like toluene was reduced in LPG. Please verify, if this was meant, as usually most significant toluene emission sources are solvent and traffic exhaust related emissions. What about other aromatics apart from toluene?*

**Response:**

Sorry for the misleading. The reduction of toluene was due to the control of solvent usage, and the increase of alkanes was due to the LPG usage.

*Revision in the manuscript:*

*Page 8, Line 31: "...such as the reduction of toluene by solvent usage control and the increase of alkanes in Liquefied Petroleum Gas (LPG) in 2005-2013 (Ou et al., 2015) and the decrease of LPG-alkanes in 2013-2014 (Lyu et al., 2016)."*

**Comment 2.20**

*Page 8 L8-10: Given the fact that TVOCs almost remained unchanged it is not that much surprising to see $O_3$ increases in VOC limited areas. It would be different in $NO_x$ limited regimes.*

**Response:**

We agree with the reviewer. This sentence and the one after have been removed from the text.

**Comment 2.21**

*Page 8 L9: It sounds like the references cited here were the first to find that urban areas in general are VOC limited. Please verify whether this is true or whether this statement refers to recent findings in Chinese cities only.*

**Response:**

5     Sorry for the misleading. That the precursor - $O_3$ relationship in urban areas are generally VOC limited has been well documented before 1990s. The references cited here verified the phenomena in Hong Kong. It is noteworthy that the related sentence has been removed for other reasons (see Response to Comment 2.20).

**Comment 2.22**

*Page 12 L11: I was just wondering if the definition of daytime (0700-1900 LT) is valid regardless what season is concerned.*

10     **Response:**

Considering weak photochemical reactions in the early morning and in the late afternoon, as well as the convenience in data calculation, the daytime period was arbitrarily defined as 0700-1900 LT. The authors admit that the duration of daytime varies by seasons, and this approach would add some uncertainty into the modelling results. However, the change in daytime duration is about 1-2 hours in Hong Kong (see Hong Kong Observation). The expected uncertainty from this would be

15     limited.

**Comment 2.23**

*Page 12 L15-16: While this statement is true for the modelled data, the observations show a completely different result.*

**Response:**

In this study, it was assumed that the $O_3$ observation included both locally-produced $O_3$ and regional transported $O_3$. So it is

20     not necessary for the modelled data (the simulated locally-produced $O_3$) and the observations to have a similar trend.

**Comment 2.24**

*Page 12 L28: "...who attributed the increasing $O_3$ trend ....to local contribution and regional transport". Isn't this statement always and at any given site true?*

**Response:**

Sorry for the misleading. The sentence has been revised accordingly.

*Revision in the manuscript:*

*Page 12, Line 33: "The spring pattern of $O_3$ in this study is consistent with the findings of Li et al. (2014) who reported the increasing $O_3$ trend (2.0 ppbv/yr) in spring at urban clusters of PRD from 2006 to 2011."*

**Comment 2.25**

*Page 12 L31-32: "... (0.70±0.34 ppbv/yr and 0.66±0.41 ppbv/yr)"... I do not see any of these values in Fig 4.*

**Response:**

Sorry for this error. The values in Figure 4 are correct. This inconsistence problem has been resolved.

*Revision in the manuscript:*

*Page 13, Line 4: "Unlike in spring, though the observed and locally-produced $O_3$ (filtered) displayed increasing trends in summer (0.67±0.34 ppbv/yr and 0.61±0.41 ppbv/yr, respectively; p<0.05)..."*

**Comment 2.26**

*Page 12 L33-34: I am astonished to read that the model did not consider the influence of solar radiation. Isn't this a crucial parameter which has not been considered?*

**Response:**

This sentence was misleading. Actually we used the TUV model to simulate the diurnal variation of solar radiation and then calculate the photolysis rate of different chemicals (see Section 2.3). However, the annual variations of solar radiation over the years were not considered in the model.

*Revision in the manuscript:*

*Page 13, Line 6: "Note that the influence of annual variation in solar radiation over the 10 years was not considered while the TUV model was used to calculate the photolysis rates, which could mask the actual trends of O3 mixing ratios."*

**Comment 2.27**

*Page 13 L1: What is the reason for the increase in solar radiation over the last years?*

**Response:**

The statement on the increase of solar radiation in summer along the 10 years was based on measurements from Hong Kong Observation, but frankly speaking, the reasons remain unknown. It might be due to less haze as the air quality has been getting better in Hong Kong and the PRD. To make this point clear, the text has been improved accordingly.

*Revision in the manuscript:*

*Page 13, Line 8: "Indeed, the total solar radiation ($0.24\pm0.16$ MJ $m^{-2}$ $yr^{-1}$, p<0.01) and temperature ($0.095\pm0.034$ K $yr^{-1}$, p<0.05) in summer significantly increased during the past 10 years (see Figure S9), subsequently resulting in the enhanced in-situ photochemical reactivity of VOCs, although their quantitative contributions remain unknown and require further investigation. The increase of solar radiation might be due to the decreasing haze as the air quality has been getting better in Hong Kong and the PRD (Louie et al., 2013)."*

**Response:**

The RIR value only shows the sensitivity of cutting precursors to $O_3$ formation and has been widely used to determine the

20 future reduction plan of precursors. However, it is not necessary for RIR to reflect the efficiency of $O_3$ production.

**Comment 2.31**

*Page 19, L3-6: It looks like RIR values for BVOC (here only isoprene) in summer are higher than those for the remaining AVOCs for the same season. Would this mean that summertime O3 production critically depends on biogenic emissions in Hong Kong in summer?*

**Response:**

Again, that RIR values for BVOC are higher than those for the remaining AVOCs in summer only indicates that $O_3$ production is more sensitive to the change in isoprene than the change in other AVOC species, from which one cannot directly derive that $O_3$ production largely depends on isoprene.

**Comment 2.32**

*Page 19, L6-7: "The higher RIR of BVOCs in summer was due to the higher photochemical reactivity". Wouldn't it be higher biogenic emissions which cause higher BVOC RIR in summer?*

**Response:**

The authors admit that, in addition to the higher photochemical reactivity, the higher biogenic emissions would be one of reasons for the higher BVOC RIR in summer.

*Revision in the manuscript:*

*Page 19, Line 15: "The higher RIR of BVOCs in summer was mainly due to the higher biogenic emissions in summer. In addition, higher photochemical reactivity of BVOCs also contributed to higher RIR of BVOCs."*

**Comment 2.33**

*Page 19, L8-10: $NO_x$ RIR is less low in winter compared to spring and summer. Wouldn't this already lead to higher $O_3$ production in winter according to the authors?*

**Response:**

The fact of less low $NO_x$ RIR does not warrant higher $O_3$ production in winter. Please refer to the Response to Comment 2.30 for reasons.

**Comment 2.34**

*Page 19, L21: Is this statement valid for entire Hong Kong or just for the TC site?*

**Response:**

Although the relationship between $O_3$ and precursors may vary by locations across Hong Kong, TC is a good representative

5  site for the overall Hong Kong (please see the Response to Comment 2.2 above). Therefore, we think this statement would

be generally valid for entire Hong Kong. Some specific locations, like those close to traffic sources, may need individual

investigation, but that is out of the scope of this study.

**Comment 2.35**

*Page 20, L9-10, Fig. 9: The trend analysis in Fig 9 is mostly driven by a few strong peaks. How robust is this analysis?*

10  *Looking into the different y-scales of Fig. 9 I conclude that summertime $O_3$ mixing ratios are largely due to the high BVOC*

*levels, which would be in line with Fig. 8. Again, is Hong Kong's $O_3$ pollution mostly caused by BVOCs?*

**Response:**

In terms of robustness of this analysis, please refer to the Response to Comment 2.1. For the contribution of BVOCs to $O_3$

formation in summer, we agree with the reviewer's conclusion, that is, summertime $O_3$ mixing ratios are largely due to the

15  high BVOC levels, given that Hong Kong has high BVOC emission in summer and low anthropogenic VOC levels due to

the dilution by clean marine air mass. However, we do not think that Hong Kong's $O_3$ pollution is mostly caused by BVOCs.

As reported by many previous studies (Zhang *et al*., 2012; Ou *et al*., 2015), anthropogenic VOCs, particularly reactive

aromatics, play a critical role in $O_3$ formation in Hong Kong and in PRD region.

**Response:**

There is a fact that alkanes include a bunch of compounds which have different but generally low reactivity with OH radicals. In this study, although the level of total alkanes increased over the years, it did not warrant the increase in its contribution to $O_3$ formation. For example, one possible case is that some alkanes with relatively high reactivity decreased with an increase of some low-reactivity alkanes.

**Comment 2.39**

*Page 21, L29-30: "In addition...blur the trend". I do not understand this sentence. Also, what photolysis rates of alkanes do the authors exactly mean?*

**Response:**

Sorry for the confusing. What we attempted to say is the reaction rate of alkanes with OH radicals.

*Revision in the manuscript:*

*Page 22, Line 2: "In addition, the seasonal variation of $O_3$ formation, of which the reaction rates of alkanes with OH radicals were high in summer and low in winter, would also blur the trend".*

**Comment 2.40**

*Page 21, L34: Here you should add the Reiman et al paper, as this was one of the first to observe anthropogenic isoprene emissions.*

**Response:**

The reviewer's advice is appreciated and the suggested reference has been added in the main text.

*Revision in the manuscript:*

*Page 22, Line 6: "The main known sources of isoprene are biogenic and anthropogenic (Borbon et al. 2001; Barletta et al., 2002; Reiman et al., 2000)."*

10    **Comment 2.42**

*Figure 4: (1) There is no gold line (ΔO₃) for summer. (2) The observations (blue line) is always the highest. What is the model missing?*

**Response:**

The regional $O_3$ ($\Delta O_3$), the difference between overall observed and simulated $O_3$, was not shown for summer in Figure 15   4, because of a negligible impact of regional transport on summer $O_3$ trend in 2005-2014.  It is evidenced by that the trend of locally-produced $O_3$ (filtered) was comparable to those of observed $O_3$ ($p$=0.12) and locally-produced $O_3$ (simulated) ($p$=0.32), respectively. To make this point clear, an explanation has been added to the caption of Figure 4.

In this study, it was assumed that the $O_3$ observation includes both local-produced and regional transported $O_3$. Thus it is not surprising to see that the observation is always higher. In addition, the regional transported $O_3$ was missing in the model 20   as it was only used to simulate the locally-produced $O_3$.

*Revision in the manuscript:*

*Caption for Figure 4: "…The regional $O_3$ in summer was negligible and is not shown in the graph."*

**Comment 2.43**

*Figure 8: Is this data day- or night-time data or both and why did the authors choose that specific time period?*

**Response:**

Only daytime data are used in Figure 8 as the RIR values were calculated based on $O_3$ production in daytime. This information has been added into the description of RIR in Section 2.3.

***Revision in the manuscript:***

*Page 7, Line 6: "The RIR is defined as the percent change in daytime $O_3$ production per percent change in precursors."*

**Comment 2.44**

*Figure S9: I am not sure about the units (M m$^{-2}$) for Solar Radiation here.*

**Response:**

Thanks for pointing out this typo which has been corrected to "MJ m$^{-2}$".

[revised manuscript text omitted]

Note: Online VOC data was available at five sites from April 2011 to January 2012; Only four months trace gases data at HKUST site were available in 2011; Totally, the simulation days at TC, YL, MK and UST are 286, 273, 233 and 105 days.

**Table S6.** The difference of locally-produced $O_3$ (simulated) before and after removing regional background VOCs.

| | HKUST (Suburban site) | Tung Chung (Suburban site) | Yuen Long (Urban site) | Mong Kok (Roadside site) |
|---|---|---|---|---|
| **Spring** | | -2.6% | -2.6% | -1.2% |
| **Summer** | -2.4% | -2.7% | -2.1% | -0.3% |
| **Autumn** | -1.8% | -1.8% | -4.0% | -3.0% |
| **Winter** | -0.6% | -4.0% | -4.8% | -4.5% |

Note: Online VOC data was available at five sites from April 2011 to January 2012; Only four months trace gases data at HKUST site were available in 2011; Totally, the simulation days at TC, YL, MK and UST are 286, 273, 233 and 105 days.

**Table S7.** The difference of locally-produced $O_3$ (simulated) before and after removing both regional background $NO_2$ and VOCs.

| | HKUST (Suburban site) | Tung Chung (Suburban site) | Yuen Long (Urban site) | Mong Kok (Roadside site) |
|---|---|---|---|---|
| **Spring** | | -33.5% | -33.2% | -12.6% |
| **Summer** | -9.8% | -13.7% | -7.9% | -2.7% |
| **Autumn** | -15.5% | -17.8% | -14% | -6.0% |
| **Winter** | -27.1% | -46.2% | -48.4% | -22.1% |

Note: Online VOC data was available at five sites from April 2011 to January 2012; Only four months trace gases data at HKUST site were available in 2011; Totally, the simulation days at TC, YL, MK and UST are 286, 273, 233 and 105 days.

**Table S8.** Seasonal averages of TVOC ($C_3$-$C_8$) in 2013 at five sites in Hong Kong.

| | Hok Tsui (Background site) | HKUST (Suburban site) | Tung Chung (Suburban site) | Yuen Long (Urban site) | Mong Kok (Roadside site) |
|---|---|---|---|---|---|
| **Spring** | 5.9±0.2 | 2.6±0.2 | 4.1±0.2 | 6.3±0.4 | 11±0.5 |
| **Summer** | 4.5±0.2 | 1.5±0.2 | 2.5±0.2 | 2.9±0.2 | 11±0.3 |
| **Autumn** | 7.6±0.2 | 2.7±0.1 | 4.7±0.2 | 7.3±0.3 | 16.1±0.4 |
| **Winter** | 13.6±0.4 | 6.7±0.3 | 7±0.3 | 16±0.7 | 24.6±0.7 |

Note: Hok Tsui is a well-known regional background site at the southeastern tip of Hong Kong. The Hong Kong University of Science and Technology (HKUST) is an Air Quality Research Supersite located on the shorefront of the HKUST campus in the Hong Kong suburban area. Yuen Long (YL) is a typical urban site adjacent to main traffic roads and surrounded by
10   residential and industrial blocks. Mong Kok is a typical roadside site with high traffic density

**Table S9.** Index of agreement (IOA) between simulated and observed $O_3$.

| Year | 2005 | 2006 | 2007 | 2008 | 2009 | 2010 | 2011 | 2012 | 2013 | 2014 |
|------|------|------|------|------|------|------|------|------|------|------|
| IOA | 0.81 | 0.82 | NA* | 0.84 | 0.77 | 0.71 | 0.77 | 0.88 | 0.79 | 0.89 |

*VOCs data in 2007 are not available due to the maintenance of the instrument.

**Table S10.** Rate constants for reactions of OH with individual VOCs measured in this study (Atkinson and Arey, 2003).

| Name | Reactivity ($10^{12} k_{OH}$) (cm$^{-3}$ molecule$^{-1}$ s$^{-1}$) | Name | Reactivity ($10^{12} k_{OH}$) (cm$^{-3}$ molecule$^{-1}$ s$^{-1}$) |
|------|------|------|------|
| Propane | 1.15 | *i*-Hexane | 5.6 |
| Propene | 26.3 | Benzene | 1.23 |
| *n*-Butane | 2.54 | *n*-Heptane | 7.15 |
| *i*-Butane | 2.34 | Toluene | 6 |
| *trans*-2-Butene | 64 | Ethylbenzene | 7.1 |
| *cis*-2-Butene | 56.4 | *m*-Xylene | 23.1 |
| *n*-Pentane | 3.94 | *p*-Xylene | 14.3 |
| *i*-Pentane | 3.9 | *o*-Xylene | 13.7 |
| 1,3-Butadiene | 66.6 | 1,3,5-trimethylbenzene | 57.25 |
| Isoprene | 101 | 1,2,4-trimethylbenzene | 32.5 |
| n-Hexane | 5.61 | 1,2,3-trimethylbenzene | 32.5 |

[Figure]

**Figure S1.** Wind rose plots showing prevailing winds in Hong Kong for each season. The daily data of 2005-2014 is
obtained from the Hong Kong airport (https://www.wunderground.com). Colour indicates wind speed (unit: km h$^{-1}$).

[Figure]

**Figure S2.** Comparison of $O_3$ production simulated by with (blue line) or without (red line) a 4-day "spin-up" time in different seasons. In the tests, four days data were randomly selected from each season in 2012 as an example. The results demonstrate that the method without the "spin-up" time indeed causes slight underestimation of the $O_3$ production with a maximum of 4.63% in spring and a minimum of 1.52% in winter.

[Figure]

**Figure S3.** Comparison of RIR calculated by with (solid bar) or without (light bar) a 4-day "spin-up" time in different seasons. In the tests, one-day data was randomly selected from each season in 2012 as an example. It shows that the approach (without spin-up time) systematically underestimates the RIR values, but it does not change the results in each season. In addition, it seems that the RIR of TVOCs is more sensitive to this method.

[Figure]

**Figure S4.** Comparison of locally-produced $O_3$ (simulated) at TC site in spring and winter with input of varying initial concentrations of $NO_2$ by subtracting the mean background $NO_2$ (red line), the mean background $NO_2$ plus 95% confidence intervals (C.I, blue line), and the mean background $NO_2$ minus 95% C.I (green line).

[Figure]

**Figure S5.** Comparison of locally-produced $O_3$ (simulated) at TC site over the year with input of varying initial concentrations of VOCs by subtracting the mean background VOCs (red line), the mean background VOCs plus 95% confidence intervals (C.I, blue line), and the mean background VOCs minus 95% C.I (green line).

[Figure]

**Figure S6.** Annual trends of simulated local $O_3$ production (blue line: with background precursors; red line: without background precursors) in four seasons at TC during 2005–2014. Error bars represent 95% confidence interval of the averages.

[revised manuscript text omitted]

---

## Referee Report (RR1)

The quality of the paper has improved significantly. However there are still a few issues which the authors want to address before publication in ACP. Here I only repeat my initial questions and include additional comments ("New"):

Major issues:

1) In most figures intra- and inter-annual variations are significantly larger than the 2005-2014 trend. For instance, the O3 trend shows the highest increase from 2005-2014 (0.67 ppb/yr) in autumn (Fig. 3). However, this trend is only determined by 3 "outlier" months of the years 2012, 2013, 2014. These 3 months are just 10% of this specific data set. Another example is Fig. 5 which shows very large scatter in O3 data for the autumn season. Also, for instance Fig. S10 about the annual trends of VOC/NOx ratios is largely determined by the last two years. The question is: how robust are all the trends shown in this paper?

New:
I want to state that the term "outliers" I used in my initial comment does not refer to any characteristics of this data, which could be regarded as "bad" data, and I understand that the data shown in box-whisker plots consists of a larger data set. This is why I put the term "outlier" in brackets. My statement focused on the fact that these few years at the end of the reporting period 2005-2014 completely determine the statistical trend and one should be cautious to deduce a trend and this should be mentioned in the text. Looking into the VOCs/NOx ratio time series (previously S10, now S17), for instance, it seems that there has been some significant variability. For instance the years 2009, 2011, and 2012 show significantly enhanced VOCs/NOx ratio. Assuming the year 2015, just following the 2005-2014 time series, would show any values comparable to those seen in 2009, 2011 and 2012, the trend analysis would yield a completely different results. The similar comment would refer to O3 time series (now S1). The authors should make the point why they believe that such variability could be excluded for future years and that their trend analysis is justified in this sense.

Other comments:

Page 4 L13-14:
How was the accuracy of 1-7% determined?

New:
I assume that a "measured mixing ratio" would ultimately depend on the weekly span checks and calibration. What makes a "measured mixing ratio" different from an "actual mixing ratio"?

Page 5 L25-29:

The authors only measured 21 VOCs. What assumptions did the authors have on other VOCs not measured, but needed as an input for MCM?

New:

The authors should include their comment specifically with respect to MCM in their manuscript.

Page 7, L16:

With regard to the precursors NOx, total VOCs and CO did the authors calculate arithmetic means or medians? Would there be differences?

New:

I primarily referred to O3 precursors, not O3 itself. I doubt that the number of data determines, whether data ensembles are distributed normally or not. Rather it is an intrinsic characteristic of the specific data ensemble. The authors should mention in the manuscript that they used arithmetic means in order to compare with other studies. Unfortunately, it does not make the use of arithmetic means more correct, the more studies have used this quantity.

Page 12 L11:

I was just wondering if the definition of daytime (0700-1900 LT) is valid regardless what season is concerned.

New:

The authors should include their answer in the manuscript.

Page 21, L17:

"...increased emissions of alkenes from traffic related sources". Is this due to enhanced alkene emissions from changes in the composition of the traffic fleet or from increased traffic volume? If it is the latter, then emissions of aromatic compounds would also increase.

New:

The authors should include their answer in the manuscript.

Page 21, L20-21:

Diesel driven vehicles emit significantly less VOCs than gasoline driven vehicles. In other words was the DCV program a significant contribution to the overall traffic related alkene emissions?

New:

The authors should include their answer in the manuscript.

Page 21, L26-28:

Why would the AVOC (alkane) contribution to O3 formation not increase with increasing alkane levels in 2005-2013?

New:

The authors should include their answer in the manuscript.

Page 22, L6-7:

The authors state that 90% of isoprene was emitted from biogenic sources, while traffic sources were less than 5%. From what sources did the remaining 5% isoprene come from?

New:

The authors should include their answer in the manuscript.

---

## Author Response (AR2)

**Responses to Referee 2's new comments**

**General comment**

The quality of the paper has improved significantly. However there are still a few issues which the authors want to address before publication in ACP. Here I only repeat my initial questions and include additional comments ("New"):

5 We highly appreciate the reviewer for reviewing our responses and the revised manuscript again. All comments have been addressed with proper revision in the manuscript. In the following section, the author's responses (in blue) are immediately after the reviewer's comments (in black for original and in green for new), with the changes in manuscript at the end (*in italic*).

**10 Major issues:**

**Comment 2.1**

In most figures intra- and inter-annual variations are significantly larger than the 2005-2014 trend. For instance, the  $O_3$  trend shows the highest increase from 2005-2014 (0.67 ppb/yr) in autumn (Fig. 3). However, this trend is only determined by 3 "outlier" months of the years 2012, 2013, 2014. These 3 months are just 10% of this specific data set. Another example is

15 Fig. 5 which shows very large scatter in  $O_3$  data for the autumn season. Also, for instance Fig. S10 about the annual trends of VOC/NOx ratios is largely determined by the last two years. The question is: how robust are all the trends shown in this paper?

New:

I want to state that the term "outliers" I used in my initial comment does not refer to any characteristics of this data, which 20 could be regarded as "bad" data, and I understand that the data shown in box-whisker plots consists of a larger data set. This is why I put the term "outlier" in brackets. My statement focused on the fact that these few years at the end of the reporting period 2005-2014 completely determine the statistical trend and one should be cautious to deduce a trend and this should be mentioned in the text. Looking into the VOCs/NOx ratio time series (previously S10, now S17), for instance, it seems that there has been some significant variability. For instance the years 2009, 2011, and 2012 show significantly enhanced VOCs/NOx ratio. Assuming the year 2015, just following the 2005-2014 time series, would show any values comparable to those seen in 2009, 2011 and 2012, the trend analysis would yield a completely different results. The similar comment would refer to O3 time series (Figure 2). The authors should make the point why they believe that such variability could be excluded for future years and that their trend analysis is justified in this sense.

**5 **Response:**

New:

10

Thanks for further clarifying the term "outlier" and giving the constructive suggestion on the issue. Indeed, the "outliers" in Figure 3 (i.e., the extremely high monthly  $O_3$  mixing ratios observed in Octobers in 2012-2014) seemed to bias the  $O_3$  trend in autumn. However, if we look at Figure S10 which showed the daily average  $O_3$  values in autumn (i.e., more data points than in Figure 3), the values varied much more significantly in 2012-2014 than in previous years. The  $O_3$  trend was determined by all the measured data points including both extremely high and low values in all the study years. We have made this point in the revised manuscript.

For VOCs/NOx ratios, we fully agree that the significant variability cannot be excluded for future in Hong Kong given the changing contributions of local emission and regional transport to these  $O_3$  precursors. Since the decreasing rate of the

15 ratios was very small, -0.02 yr-1 (Figure S17), over-interpretation of the result should be avoided despite the result is statistically significant (p

**Revision in the manuscript:**

20 Page 11, line 16:

25

"It is noteworthy that there were three extremely high  $O_3$  data points in Octobers in 2012-2014 (Figure 3), which seemed to bias the  $O_3$  trend in autumn. However, if we looked at Figure S10 which showed the daily average  $O_3$  values in autumn (i.e., more data points than in Figure 3), the values varied much more significantly in 2012-2014 than in previous years. It is worth emphasizing that the overall  $O_3$  trend was determined by all the measured data points including both extremely high and low values in all the study years." Page 20, line 32:

5

"Furthermore, the monthly variation of TVOC/NOx ratios showed a statistically significant decreasing trend at a rate of -0.02 yr-1 (p<0.05) (see Figure S17). The weak declining trend moderately supports that VOC reduction became more effective in reducing O3 in the past 10 years, which is consistent with the conclusions from the above modelling results."

Supplementary Figure S17:

To facilitate the discussion above, the format of Figure S17 has been changed from a bar chart to a time-series plot with dots.

10 Figure S17. Monthly trend of TVOCs/NOx ratio at TC in 2005-2014.

Minor issues:

**Comment 2.2**

Page 4 L13-14:

How was the accuracy of 1-7% determined?

New:

5 I assume that a "measured mixing ratio" would ultimately depend on the weekly span checks and calibration. What makes a "measured mixing ratio" different from an "actual mixing ratio"?

**Response:**

New:

The weekly span checks and calibration are part of the routine QA/QC to improve the accuracy of VOC measurements.

10 However, many other known or unknown sources, including at least but not limited to the preparation and storage of the standard gases, the variation of the instrumental performance, the operators and so on, making it impossible to eliminate the difference between "measured mixing ratio" and an "actual mixing ratio".

As the description of measurement techniques and related QA/QC has been greatly enhanced in previous revision, no more revision has been made for this comment.

**15 Comment 2.3**

Page 5 L25-29:

The authors only measured 21 VOCs. What assumptions did the authors have on other VOCs not measured, but needed as an input for MCM?

New:

20 The authors should include their comment specifically with respect to MCM in their manuscript.

**Response:**

Thanks for the comment. We have included our comment specifically with respect to MCM in this version as follows.

**Revision in the manuscript:**

Page 6, Line 3:

"It was assumed that the measured VOCs contributed a dominant fraction to  $O_3$  production, and that the initial concentrations of those VOCs not measured but needed by MCM were zero. We admit that the use of a limited number of VOCs would cause photochemical reactivity missing."

**Comment 2.4**

**5 Page 7, L16:**

With regard to the precursors NOx, total VOCs and CO did the authors calculate arithmetic means or medians? Would there be differences?

New:

I primarily referred to  $O_3$  precursors, not  $O_3$  itself. I doubt that the number of data determines, whether data ensembles are

10 distributed normally or not. Rather it is an intrinsic characteristic of the specific data ensemble. The authors should mention in the manuscript that they used arithmetic means in order to compare with other studies. Unfortunately, it does not make the use of arithmetic means more correct, the more studies have used this quantity.

**Response:**

New:

15 Thanks for the clarification. The authors fully understand the reviewer's concern. The suggested information has been added in the text.

**Revision in the manuscript:**

Page 8, Line 12:

"Please note that arithmetic means were used here in order to compare with other studies."

**20 Comment 2.5**

Page 12 L11:

I was just wondering if the definition of daytime (0700-1900 LT) is valid regardless what season is concerned.

New:

The authors should include their answer in the manuscript.

**Response:**

The answers have been included in the revised manuscript.

**Revision in the manuscript:**

5 Page 13, Line 17:

"Although the actual duration of daytime in Hong Kong varies in 1-2 hours by seasons, the expected uncertainty from it would be limited if considering weak photochemical reactions in early morning and in late afternoon."

**Comment 2.6**

Page 21, L17:

10 "...increased emissions of alkenes from traffic related sources". Is this due to enhanced alkene emissions from changes in the composition of the traffic fleet or from increased traffic volume? If it is the latter, then emissions of aromatic compounds would also increase.

**New:**

The authors should include their answer in the manuscript.

**15 **Response:**

The answers have been included in the revised manuscript.

**Revision in the manuscript:**

Page 22, Line 23:

"In contrast, the contributions of AVOC (alkenes) to  $O_3$  production in these years showed a significantly increasing trend

20 with a rate of 0.14±0.01 ppbv/yr (p<0.05), perhaps attributed to the increased emissions of alkenes from changes in the composition of the traffic fleet and from increased traffic volume."

**Comment 2.7**

Page 21, L20-21:

Diesel driven vehicles emit significantly less VOCs than gasoline driven vehicles. In other words was the DCV program a significant contribution to the overall traffic related alkene emissions?

**New:**

The authors should include their answer in the manuscript.

**5 **Response:**

The answers have been included in the revised manuscript.

**Revision in the manuscript:**

Page 22, Line 27:

"Among the measures, gasoline and LPG vehicular emissions caused ambient alkenes to increase during the same period

10 due to the increasing number of LPG/gasoline vehicles and some short-term/non-mandatory measures (Lyu et al., 2017). The diesel commercial vehicle (DCV) program (2007-2013) was shown to be effective in reducing the emission of alkenes from diesel vehicles (Lyu et al., 2017), however these vehicles emit significantly less VOCs than gasoline driven vehicles. In consequence, the overall emissions of alkenes from traffic related sources increased during 2005-2014, leading to the increased contribution of AVOC (alkenes) to O3 formation (Lyu et al., 2017)."

15

**Comment 2.8**

Page 21, L26-28:

Why would the AVOC (alkanes) contribution to  $O_3$  formation not increase with increasing alkane levels in 2005-2013? New:

20 The authors should include their answer in the manuscript.

**Response:**

The answers have been included in the revised manuscript.

**Revision in the manuscript:**

Page 23, Line 1:

"Unlike AVOC (Aromatics/Alkenes), the contribution of AVOC (Alkanes) to  $O_3$  formation during 2005-2014 showed no significant change (p=0.23) despite the increase of total alkane levels in the atmosphere in 2005-2013 (Ou et al., 2015). This is because alkanes include a bunch of compounds which have different but generally low reactivity with OH radicals. Hence, although the level of total alkanes increased over the years, it did not warrant the increase in its contribution to  $O_3$

5 formation. For example, one possible case is that some alkanes with relatively high reactivity decreased with an increase of some low-reactivity alkanes."

**Comment 2.9**

Page 22, L6-7:

The authors state that 90% of isoprene was emitted from biogenic sources, while traffic sources were less than 5%. From

10 what sources did the remaining 5% isoprene come from?

**New:**

The authors should include their answer in the manuscript.

**Response:**

The answers have been included in the revised manuscript.

**15 *Revision in the manuscript:**

Page 23, Line 15:

"The source of isoprene at TC site has been also investigated and confirmed by previous long-term source apportionment studies, which reported that during 2005-2013 about 90% of isoprene was emitted from biogenic emissions, with minor contribution from traffic emission, consumer products and printing processes (Ou et al., 2015)."